# The in vivo genetic program of murine primordial lung epithelial progenitors

Laertis Ikonomou [1,2]*, Michael J. Herriges[1,2], Sara L. Lewandowski [1,2], Robert Marsland III [3], Carlos Villacorta-Martin[1], Ignacio S. Caballero[1], David B. Frank[4], Reeti M. Sanghrajka[1,2], Keri Dame[1,2], Maciej M. Kańduła[5,6], Julia Hicks-Berthet [7], Matthew L. Lawton [1,2], Constantina Christodoulou[2], Attila J. Fabian[8], Eric Kolaczyk[5], Xaralabos Varelas [7], Edward E. Morrisey[9], John M. Shannon[10], Pankaj Mehta [3] & Darrell N. Kotton[1,2]*

Multipotent Nkx2-1-positive lung epithelial primordial progenitors of the foregut endoderm are thought to be the developmental precursors to all adult lung epithelial lineages. However, little is known about the global transcriptomic programs or gene networks that regulate these gateway progenitors in vivo. Here we use bulk RNA-sequencing to describe the unique genetic program of in vivo murine lung primordial progenitors and computationally identify signaling pathways, such as Wnt and Tgf-β superfamily pathways, that are involved in their cell-fate determination from pre-specified embryonic foregut. We integrate this information in computational models to generate in vitro engineered lung primordial progenitors from mouse pluripotent stem cells, improving the fidelity of the resulting cells through unbiased, easy-to-interpret similarity scores and modulation of cell culture conditions, including sub-stratum elastic modulus and extracellular matrix composition. The methodology proposed here can have wide applicability to the in vitro derivation of bona fide tissue progenitors of all germ layers.

[1] Center for Regenerative Medicine, Boston University and Boston Medical Center, Boston, MA 02118, USA. [2] The Pulmonary Center and Department of Medicine, Boston University School of Medicine, Boston, MA 02118, USA. [3] Department of Physics, Boston University, Boston, MA 02215, USA. [4] Division of Pediatric Cardiology, Department of Pediatrics, The Children's Hospital of Philadelphia, Philadelphia, PA 19104, USA. [5] Department of Mathematics & Statistics, Boston University, Boston, MA 02215, USA. [6] Chair of Bioinformatics Research Group, Boku University, 1190 Vienna, Austria. [7] Department of Biochemistry, Boston University School of Medicine, Boston, MA 02118, USA. [8] Biogen Inc., 225 Binney St, Cambridge, MA 02142, USA. [9] Penn Center for Pulmonary Biology, University of Pennsylvania, Philadelphia, PA 19104, USA. [10] Division of Pulmonary Biology, Cincinnati Children's Hospital, Cincinnati, OH 45229, USA. *email: laertis@bu.edu; dkotton@bu.edu

Pluripotent stem cell (PSC)-based systems offer the possibility of de novo somatic cell derivation through directed differentiation, a multistage process recapitulating developmental milestones[1]. This methodology relies heavily on prior knowledge of developmental pathways and processes within the tissue/organ of interest, and incomplete developmental knowledge can be a significant impediment to the establishment of efficient directed differentiation protocols[2].

In the case of lung specification, elucidation of the role of Bmp and Wnt signals through loss-of-function studies in murine development[3–5] has led to the development of growth factor cocktails for derivation of lung progenitors from mouse and human PSCs[6–9]. Similarly, the in vitro derivation of ciliated cells[10,11] as well as proximal and distal lung progenitors[12,13] were made possible by in vivo studies of Notch signaling in ciliated cell differentiation[14] and Wnt signaling in lung proximal-distal patterning[15–17], respectively.

An important intermediate population in lung lineage derivation from PSCs is the lung epithelial primordial progenitor population, i.e., the first cells within the foregut that are specified to a lung epithelial cell fate, but do not yet express markers of lung epithelial differentiation, such as surfactant genes. Nkx2-1, the earliest marker of lung fate, is first expressed in the prospective lung domain of the gut tube at around embryonic (E) day 9.0 during mouse embryogenesis[18]. It is also expressed in the prospective thyroid domain and in the developing forebrain[19]. Previous work with mouse and human PSC lines has therefore used in vitro Nkx2-1 expression as a marker of lung progenitors[6–9,12,13,20] with both distal alveolar and proximal airway epithelial competence. In addition, our previous work using Nkx2-1 fluorescent reporter lines has allowed us to purify and study endodermal Nkx2-1$^+$ populations[7,9,20,21]. Nevertheless, in the absence of similar characterization of the earliest primary Nkx2-1$^+$ progenitors that arise during mammalian organogenesis, a major question remains, how closely do the in vitro progenitors resemble their in vivo counterparts? Indeed, previous efforts to study early thyroid and lung patterning have used microdissection techniques that inevitably lead to isolation of mixed populations[22]. Furthermore, the equivalent stage in human development is not readily accessible and in vivo information on the lung primordium, even from other species, would be highly informative for human PSC-based lung differentiation studies.

Here, we report the identification, purification, and global transcriptomic analysis of the earliest detectable in vivo mouse lung progenitors. We compare the genetic program of these developing lung progenitors to foregut endodermal precursors purified just prior to the onset of expression of the Nkx2-1 program and contrast these programs to those of the Nkx2-1$^+$ lineages that compose the developing forebrain and thyroid. These comparisons delineate the unique genetic program of the lung primordium at the moment of lung lineage specification in vivo and suggest pathways that regulate the emergence of lung fate within the developing foregut endoderm. We then mine the global transcriptome of the in vivo lung primordium to improve the critical stage of in vitro lung specification in a mouse PSC-based system. We show that modification of cell culture conditions, including biomechanical properties of the culture substratum, can enhance the epithelial character of PSC-derived mouse lung progenitors. By applying computational methods for classifying cell fate consisting of Linear Algebra-based projections, we demonstrate the enhanced fidelity of progenitors derived under improved conditions as reflected by increased projection scores onto the in vivo lung epithelial primordial population. Thus, our approach delineates the unique genetic program of the earliest primordial lung progenitors and underscores the importance of applying such an in vivo roadmap to guide the specification of primordial organ progenitors that may be engineered in vitro from alternate sources, such as PSCs.

## Results

**Lung epithelium is derived from Nkx2-1+ primordial progenitors.** We first tested whether lung epithelial lineages are derived from lung primordial progenitors in mouse development. Prior reports have characterized the lung epithelia that express Nkx2-1[23] and demonstrate that Nkx2-1 lineage labeling of specific lung cell types such as pulmonary neuroendocrine cells[24] and type II alveolar epithelial cells (AECs)[25] is evident when using a transgenic Nkx2-1$^{Cre}$ driver mouse line[26] (Supplementary Fig. 1A). In our hands, this line labeled both proximal and distal lung epithelia (Supplementary Fig. 1B), as well as thyroid and forebrain cells. However, labeling in the lung did not occur until E11.5 (Supplementary Fig. 1C) making this line unsuitable for marking the progeny of E9.0 lung primordial progenitors. Next, we employed a mouse knock-in line, Nkx2-1$^{CreERT2}$ [27], and crossed it with the R26R$^{nT/nG}$ line[28] (Fig. 1a). In the resulting inducible mouse line (Nkx2-1$^{trace}$), Nkx2-1-expressing cells and their progeny should indelibly be marked by nuclear enhanced green fluorescence protein (hereafter nGFP) upon tamoxifen administration. Indeed, when timed-pregnant mice were exposed to tamoxifen before E9.0 (Fig. 1b; Supplementary Fig. 1D, E), nGFP-expressing cells throughout the developing lung epithelium were observed at E14.5 and E18.5. GFP expression was strictly localized to the epithelial compartment, as shown by co-staining with the pan-epithelial marker EPCAM (Supplementary Fig. 1E) and NKX2-1 (Fig. 1c). At E18.5, nGFP+cells were found to co-express markers of type II AECs (SFTPC), type I AECs (PDPN), basal (P63), club (SCGB1A1), and ciliated (acetylated α-tubulin) cells (Fig. 1c). Lineage labeling was observed as expected in the forebrain, but not in the liver or stomach (Supplementary Fig. 1F). Therefore, the vast majority of epithelial lung lineages derive from lung Nkx2-1$^+$ progenitors at around E9.0. We considered the possibility that ongoing lineage labeling might result from persistent circulating tamoxifen for days after the E8.75 injection time point, however, no lung lineage labeling was observed in embryos injected with tamoxifen at day E7.5 and harvested at E13.5 (Supplementary Fig. 1G).

**Faithful and specific lung expression of the Nkx2-1$^{GFP}$ reporter.** Having established that proximal and distal lung epithelia are derived through an Nkx2-1$^+$ lung epithelial progenitor intermediate, we then sought to develop an approach for the identification and isolation of Nkx2-1$^+$ primordial progenitors as they emerge from foregut endodermal precursors. We previously reported the generation of a mouse line carrying a cytoplasmic GFP reporter construct targeted to the Nkx2-1 locus (hereafter Nkx2-1$^{GFP}$) (Fig. 1d)[7], and we analyzed the kinetics of Nkx2-1$^{GFP}$ expression in this mouse from adulthood back to endodermal gut tube formation (Fig. 1e, f). Nkx2-1$^{GFP}$ expression in the prospective lung domain was first detectable at E9.0 (18–23 somites), consistent with the time of lung lineage specification. Prior to E9.0, GFP was expressed only in the thyroid and forebrain domains, consistent with earlier initiation of the specification of these Nkx2-1$^+$ lineages[29]. GFP reporter expression recapitulated endogenous Nkx2-1 transcript expression in all three embryonic domains (forebrain, thyroid, and lung) (Fig. 1d). Within the lung, GFP expression continued exclusively in the epithelium throughout development, consistent with previously described Nkx2-1 expression patterns[30]. Postnatally, GFP+cells co-expressed nuclear NKX2-1 protein and markers of either club cells (SCGB1A1), basal cells (P63), or type II AECs (SFTPC) (Fig. 1f). GFP expression in Type I AECs (PDPN) and ciliated

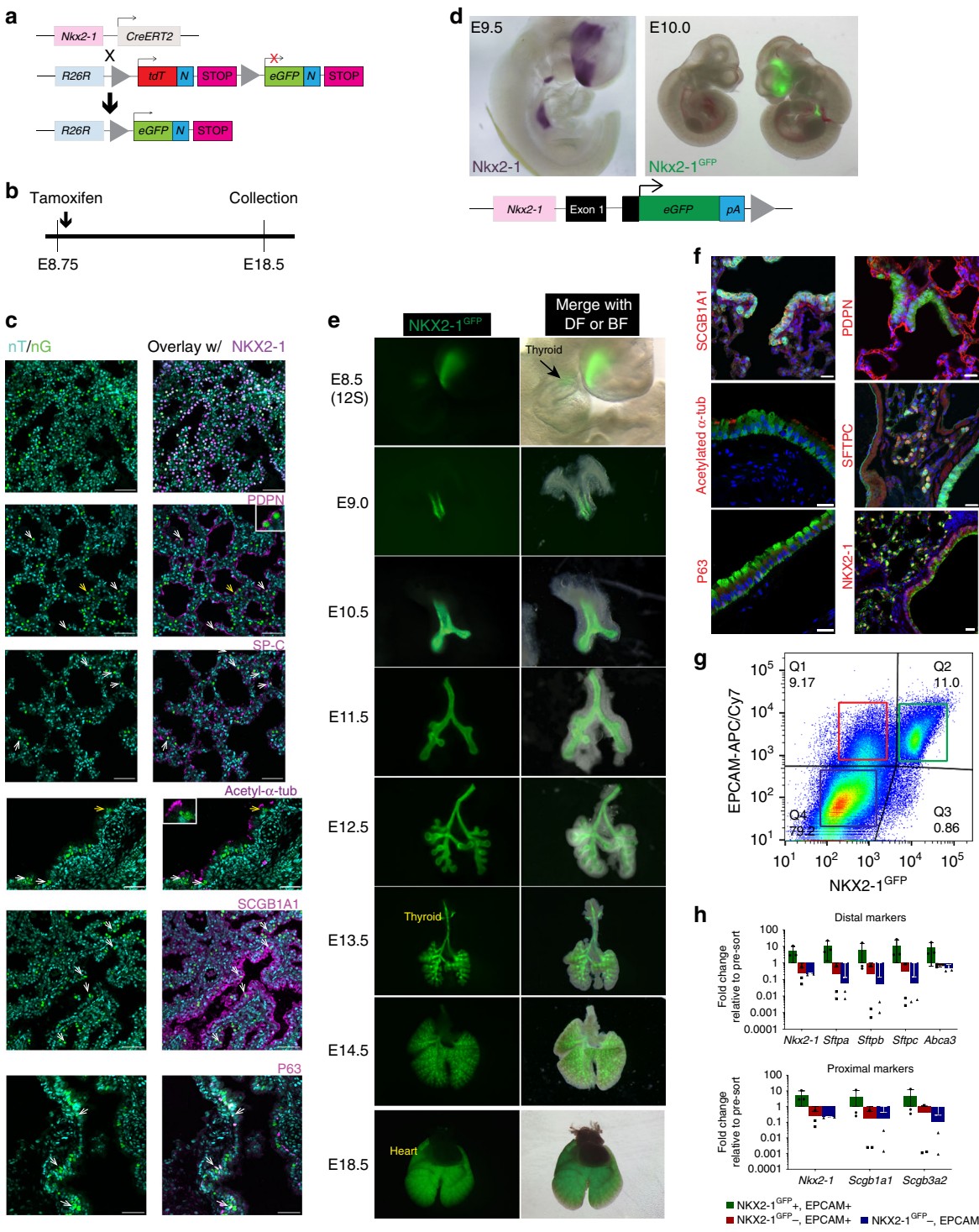

cells (acetylated α-tubulin) was more variable, with low to non-detectable GFP in these cell types (Fig. 1f). The immunostaining data were corroborated by the enrichment of club (*Scgb1a1, Scgb3a2*) and Type II AEC marker (*Sftpa, Sftpb, Sftpc,* and *Abca3*) gene expression in sorted NKX2-1$^{GFP+}$EPCAM$^+$ cells (Fig. 1g, h). These data correlated well with prior reports that have established robust expression of NKX2-1 in postnatal club cells and Type II AECs and studies that demonstrate NKX2-1 expression in both early developing P63$^+$ lung epithelia as well as postnatal mature P63$^+$ airway basal cells[31,32]. Interestingly, a recent report indicated that NKX2-1 expression is required for both Type I AEC development and cell-fate maintenance[33].

Taken together, these data indicate that GFP expression recapitulates endogenous Nkx2-1 expression faithfully and specifically in the Nkx2-1$^{GFP}$ mouse. Therefore, this mouse can be used to sort to purity lung epithelial progenitors at the moment of lung specification, E9.0.

**Distinct genetic program of lung primordial progenitors.** Next, we sought to understand the genetic programs of the lung, forebrain, and thyroid progenitors during mouse development. We purified Nkx2-1$^{GFP+}$ progenitors from the forebrain, lung, and thyroid by flow cytometry and analyzed their global

**Fig. 1 Identification and lineage tracing of Nkx2-1[+] lung epithelial cells. a** Schematic of *Nkx2-1*[CreERT2]; *R26R*[nT/nG] mice. **b** Lineage-tracing experimental design using the mice described in (**a**) that express either nuclear tdTomato (nT; turquoise pseudocolor) prior to Cre recombination vs. nuclear GFP (nG; green pseudocolor) after Cre recombination. **c** Confocal micrographs of lung paraffin tissue sections showing lineage-labeled cells co-stained with markers of lung epithelial cell types. Arrows indicate lineage-traced (ntdTomato[-]/nGFP+; nT-/nG+) cell co-localization with markers of lung epithelial cells. For acetyl-α-tubulin and PDPN stains, yellow arrows indicate region shown in inset. Representative images from three lineage-traced embryos, Scale bars: 50 μm. **d** Schematic of the *Nkx2-1*[GFP] knock-in reporter mouse (upper panel) with *Nkx2-1* ISH at E9.5 (lower left panel) and Nkx2-1[GFP] reporter expression in forebrain, thyroid and lung domains at E10.0 (lower right panel). Notice absence of Nkx2-1[GFP] expression in wild-type littermate. NB: the *nuclear* GFP lineage tracer in panels **a**–**c** (nG) is a different GFP than the knock-in *cytoplasmic* Nkx2-1[GFP] reporter shown in **d**–**g**. **e** Epifluorescence stereomicrographs of Nkx2-1[GFP] expression time course during lung development in the Nkx2-1[GFP] knock-in mouse demonstrate that the reporter is faithful and specific. Nkx2-1[GFP] thyroid is situated in front of the trachea at E13.5 (arrowhead). DF dark field, BF bright field. Representative images from embryos derived from three to ten independent litters per time point. **f** Confocal micrographs of adult Nkx2-1[GFP] mouse lung cryosections. NKX2-1[GFP] expression is evident in club (SCGB1A1), Type II alveolar epithelial (SFTPC), and basal cells (P63) but low or undetectable in ciliated (acetylated α-tubulin) and Type I alveolar epithelial cells (PDPN). The PDPN micrograph is a maximum intensity projection of six 0.82 μm optical slices. Representative images from three adult mice. Scale bars: 20 μm. **g** Bivariate flow cytometry dot plot indicating populations with various levels of NKX2-1[GFP] and EPCAM (color gates) and **h** RT-qPCR analysis of sorted populations showing enrichment of proximal and distal lung marker expression in the EPCAM[+] NKX2-1[GFP+] fraction, $N = 3$ independent sorts, error bars represent standard deviation.

transcriptomes by RNA-Sequencing (RNA-Seq) (Fig. 2a; Supplementary Fig. 2A). On average, we identified 100 GFP+primordial lung progenitors per embryo at E9.0 with higher numbers (~1300 cells) of GFP+forebrain cells accessible at this time point. Due to the low number of thyroid progenitors at E9.0 (~50 purified cells/embryo), we purified epithelial thyroid progenitors at a later time point (E13.5) in order to extract sufficient RNA for analysis (Fig. 2c). Highly enriched *Nkx2-1* expression in sorted Nkx2-1[GFP+] cells at three time points confirmed the high purity of the sorts as well as the specificity of the reporter by both RT-qPCR and RNA-Seq (Supplementary Fig. 2B; Fig. 2d, respectively). On average, forebrain cells expressed higher levels of Nkx2-1 transcripts compared to E9.0 lung and E13.5 thyroid. As two alternative *Nkx2-1* transcripts have been reported[34], one including all three exons and one including exons 2 and 3[34], we mapped *Nkx2-1* sequencing reads on the *Nkx2-1* locus (Fig. 2e). No obvious difference of transcript distribution was found between the three Nkx2-1-expressing populations.

Since Nkx2-1[+] lineages in the lung or thyroid arise from foregut endodermal precursors and Nkx2-1[+] forebrain originates from the ectodermal germ layer, we used established flow cytometry sorting algorithms[35] to prepare comparator RNA-Seq data sets from purified embryonic foregut endodermal (ENDM1[+] EPCAM[+]) and ectodermal (ENDM1[-]EPCAM[+]) populations isolated from E8.25 embryos (Fig. 2a; Supplementary Fig. 2D). Enrichment of *Sox17* expression in the ENDM1[+]EPCAM[+] fraction (Supplementary Fig. 2C), confirmed the specificity of the sort. All the populations used in RNA-Seq analysis are shown in Fig. 2b.

Lung primordial progenitors appear to possess a genetic program quite distinct from both thyroid and forebrain progenitor populations as well as foregut endoderm, their precursor population (Fig. 2f). To further understand the distinct genetic programs of embryonic Nkx2-1[+] progenitors, we performed pairwise comparisons between these progenitors and their precursor populations (foregut endoderm for lung and thyroid, ectoderm for the forebrain). As shown in Fig. 3a, different sets of genes were differentially expressed between Nkx2-1[+] progenitors and pre-specified populations (false discovery rate (FDR) < 0.05, average expression > 0 and |log₂(fold change)| > 2).

Nine hundred eighty-five genes were differentially expressed between lung primordial progenitors and foregut endoderm. Those upregulated in the lung primordium included *Nkx2-1*, transcription factors (TFs) expressed in early lung epithelium (*Foxp2*, *Irx2*)[36,37] and specific *Hox* genes (*Hoxa2*, *Hoxa4*). Definitive endoderm, primitive streak, and node marker genes

such as *T*, *Eomes*, *Sox17*, and *Nodal* were downregulated in the lung primordium, consistent with lineage specification. Of note, *Dpp4* (*Cd26*) that we have previously published as a negative selection marker of human PSC-derived lung progenitors[20] was significantly downregulated in the lung primordial cell population. Regarding the thyroid genetic program, 2340 genes were differentially expressed between E13.5 thyroid epithelium and foregut endoderm. Those upregulated in thyroid progenitors included the TFs *Nkx2-1*, *Pax8*, *Foxe1*, and *Hhex* that form an integrated network controlling thyroid cell fate[38]. Lastly, the genetic program of forebrain specification consisted of 1233 genes differentially expressed between E9.0 Nkx2-1[+] forebrain progenitors and ectoderm. Among the genes upregulated in the former were *Otx1/2*, *Dbx1*, *Six3*, *Lhx5*, which have all well-established roles in forebrain development[39–42]. Interestingly, *Foxa2* was also upregulated in the E9.0 forebrain, signifying that *Nkx2-1/Foxa2* co-expression in vitro is not always indicative of lung endodermal progenitors, as has been widely assumed.

Pairwise comparison between the two Nkx2-1[+] endodermal progenitor cell populations showed differential expression of 1257 genes (Fig. 3b). Genes upregulated in the thyroid progenitor population included *Pax8*, *Hhex*, and *Foxe1* as well as genes including *Cd44*, *Prlr*, *Chdh*, *Scara5*, *Slc4a5*, and *Cckar* that are enriched in the E10.5 thyroid bud[22]. Nkx2-1 was also upregulated, consistent with higher relative expression levels in thyroid compared to lung epithelia (Fig. 2d, e; Supplementary Fig. 2B). Wnt target genes (*Lef1* and *Nkd2*) were upregulated in the lung epithelial progenitors, reflecting the distinct role of Wnt signaling in lung vs. thyroid specification[9,21].

Having outlined the distinct temporal genetic programs of the lung, thyroid, and forebrain progenitors, we then tried to define tissue-specific transcriptomic signatures in an unbiased manner by comparing the three Nkx2-1[+] populations (false discovery rate (FDR) < 0.05, average expression > 0 and |log₂(fold change)| > 3) (Fig. 3c). We found 679 genes with significant gene expression changes, organized in three clusters, outlined in Fig. 3c. The "thyroid" cluster contained two of the key thyroid TFs (*Pax8*, *Hhex*), markers of early thyroid epithelium (*Prlr*)[9,22], as well as putative thyroid markers described by us in mouse PSC-derived thyrocytes (*Aqp1*, *Col25a1*)[43]. The "lung" cluster contained *Gata4/6*, *Irx2*, *Hoxb8*, and *Cpm*, a marker of human PSC-derived ventralized Nkx2-1[+] foregut progenitors[44]. *Ankrd1*, a canonical target of the Hippo pathway, was found in this cluster and also among the genes that distinguish the lung primordium from foregut endoderm (Fig. 3a, c). On the other hand, the "forebrain" cluster contained genes (*Dbx1*, *Otx1/2*, *Lhx5*, *Foxd1*, and *Six3*) that distinguish forebrain from pre-specified ectoderm

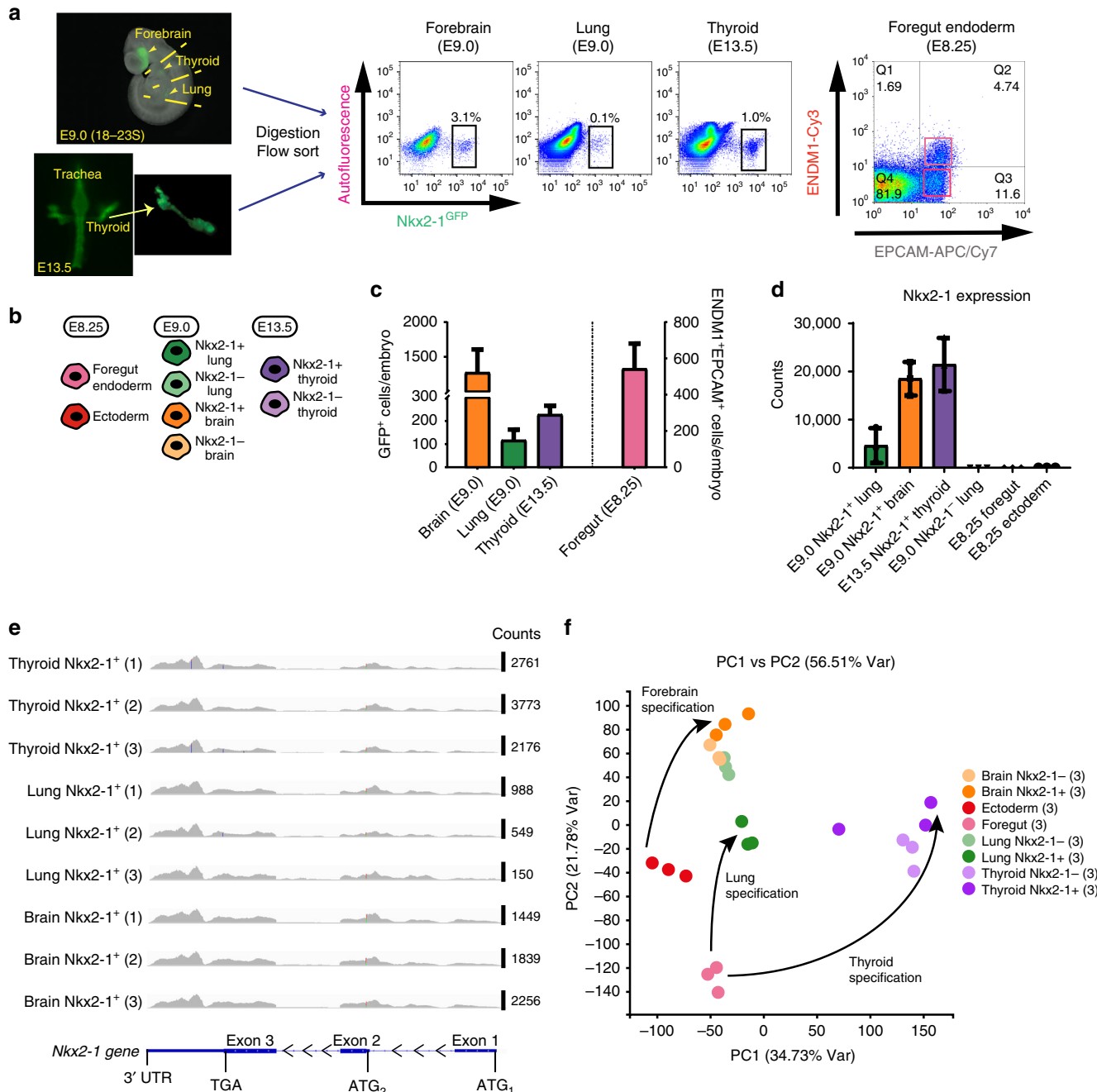

**Fig. 2 RNA-Seq analysis of purified mouse embryonic Nkx2-1+ populations. a** Schematic of embryo dissection and NKX2-1GFP+ cell sorting at the lung primordium stage (E9.0, 18–23 somites) and at E13.5. The Nkx2-1GFP+ lung, thyroid, and forebrain domains were micro-dissected using an epifluorescence stereomicroscope. At E13.5, thyroid is separated from the trachea prior to enzyme digestion and sorting (left panels). Bivariate flow cytometry dot plots showing sorted NKX2-1GFP+ cell populations (middle panel) and pre-specified foregut endoderm (ENDM1+EPCAM+) and ectoderm (ENDM1−EPCAM+) (right panel). **b** FACS-purified cell populations used in RNA-Seq analysis. The same colors are consistently used in subsequent figures to identify the respective populations. **c** The number of cells recovered by flow cytometry and normalized per embryo for the NKX2-1GFP+ populations (lung, thyroid, and forebrain) and foregut endoderm. The number of sorts: $N = 7$ for the lung, thyroid, and forebrain; $N = 5$ for foregut endoderm, error bars represent standard deviation. **d** *Nkx2-1* expression (RNA-Seq normalized counts) in sorted Nkx2-1GFP+ and Nkx2-1GFP-negative populations. **e** *Nkx2-1* counts for all *Nkx2-1*GFP+ samples ($N = 3$ lung, thyroid, and forebrain samples) mapped on the *Nkx2-1* locus. Normalized *Nkx2-1* counts for each sample are indicated on the right. **f** Principal component analysis (PCA) plot of the eight populations depicted in (**b**). Arrows connect specified endodermal or ectodermal populations and their respective precursor stages. The partial overlap of the Nkx2-1− lung and forebrain field populations is most probably due to the fact that both populations are heterogeneous and contain mesenchymal, endothelial, neuronal, non-lung foregut, and other lineages.

(Fig. 3a) and that have also been identified by us as part of the transcriptional signature of in vitro human PSC-derived NKX2-1GFP+ putative forebrain progenitors[20].

This analysis demonstrates the distinct genetic programs of Nkx2-1+ developing lineages and how these programs temporally differ from cells within each relevant germ layer.

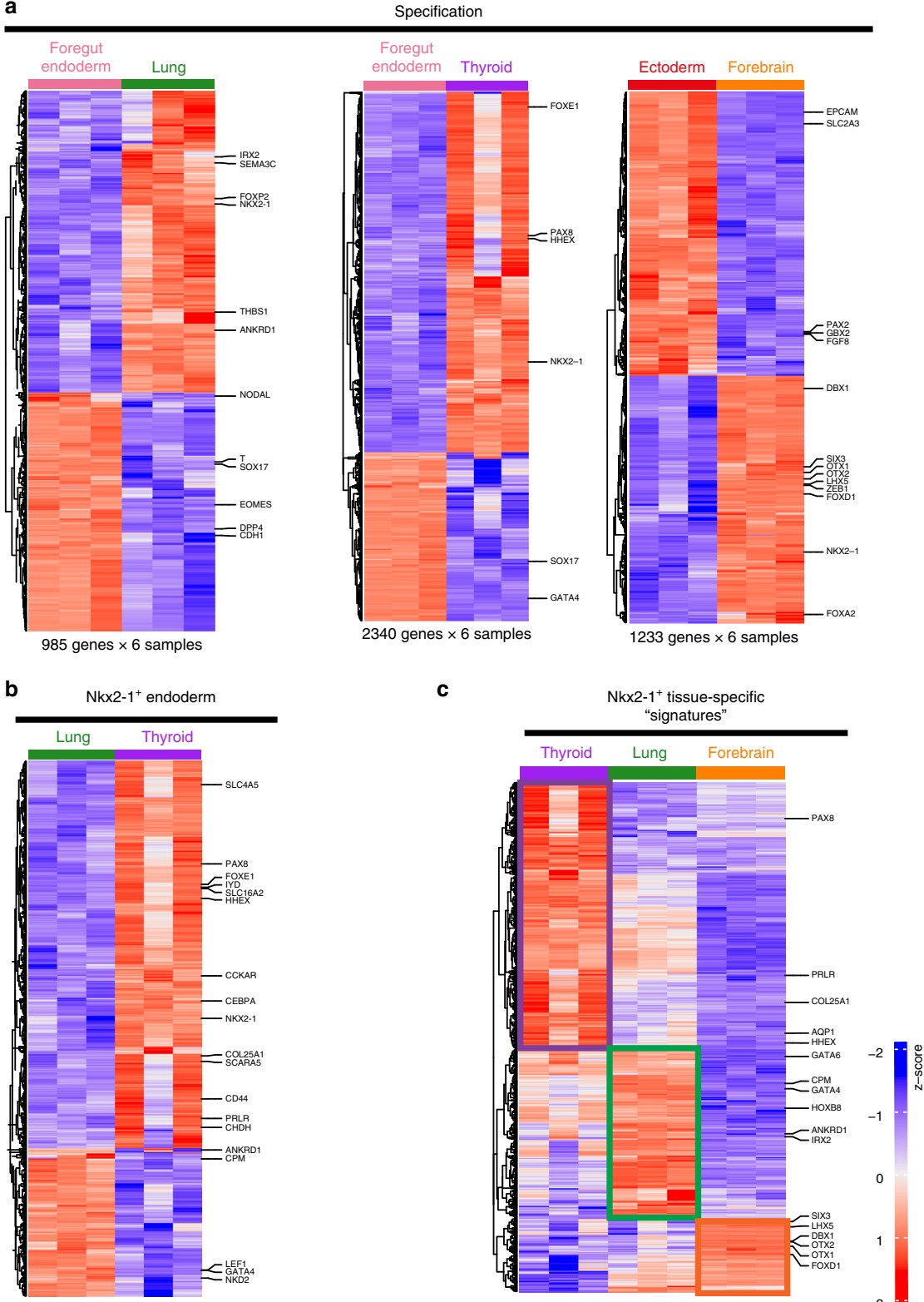

**Fig. 3 Gene expression profiles of Nkx2-1[+] endodermal and ectodermal populations. a** Heatmaps containing row-normalized z-scores and representing unsupervised hierarchical clustering of samples analyzed in population RNA-Seq. Each heatmap depicts the pairwise comparison of specified endodermal or ectodermal Nkx2-1[+] populations and their precursor stage (foregut endoderm and ectoderm, respectively). The number of genes used to create each heatmap appears underneath the heatmap. The cutoff criteria used define the number of genes for each heatmap are: false discovery rate (FDR) < 0.05, average expression > 0 and |log$_2$(fold change)| > 2. **b** Heatmap of differentially expressed transcripts between the two Nkx2-1[+] endodermal progenitor cell populations (lung and thyroid). The same cutoff criteria as in (**a**) were used. **c** Heatmap of differentially expressed transcripts across the three Nkx2-1[+] cell populations (tissue-specific gene "signatures"). The same cutoff criteria as in (**a**) were used with the only difference of |log$_2$(fold change)| > 3.

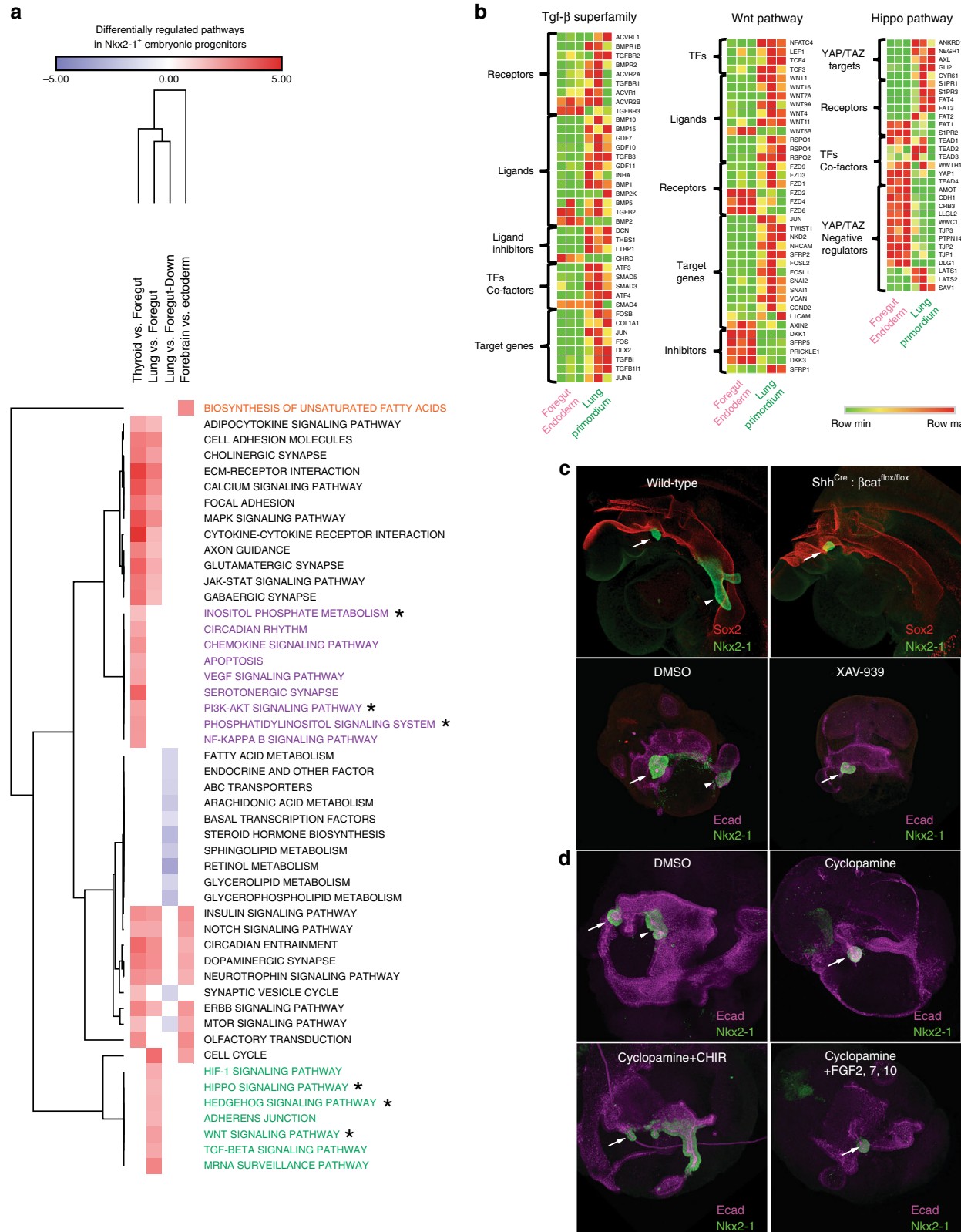

**Wnt, Hippo, and Tgf-β pathways in in vivo lung specification**. To delve into higher order biological processes that are unique to lung specification, we performed gene set enrichment analysis (GSEA) for all three Nkx2-1⁺ embryonic progenitors and their respective pre-specified populations (Fig. 4a; Supplementary Fig. 3A). Several pathways/processes appear to be common in either all three progenitor populations (e.g., "Circadian

Entrainment"), or in two out of three populations (e.g., "Cytokine–Cytokine Receptor Interaction" in the lung primordium and thyroid). On the other hand, certain signaling pathways were progenitor-specific (seven for the lung, ten for thyroid, and one for forebrain, Supplementary Fig. 3B). Wnt, Hedgehog, Tgf-β superfamily, and Hippo pathways were upregulated exclusively in lung primordial progenitors upon

**Fig. 4 Differential regulation of the Hippo, Wnt and Tgf-β superfamily pathways in lung specification. a** Heatmap of normalized enrichment scores (NES) for pathways that are differentially regulated in specified *Nkx2-1*[+] embryonic populations by gene set enrichment analysis (GSEA). The KEGG pathways database was used as basis for GSEA and an FDR cutoff of 0.25 was used for inclusion of pathways for each pairwise comparison. Pathways that were investigated are indicated with an asterisk (*). **b** Heatmaps of selected transcripts for the Hippo, Wnt, and Tgf-β superfamily pathways showing differential expression between pre-specified foregut endoderm and *Nkx2-1*[+] lung primordium. **c** Whole-mount confocal micrographs of uncultured, freshly isolated embryos at 25–28 somite stage (upper panel), or mouse foregut explants (lower panel). Foreguts were explanted at the eight-somite stage and cultured for 2 days. Nkx2-1 expression demarcates the lung and thyroid domains. Co-staining with Sox2 (upper panel) and E-cad (lower panel) is used to visualize foregut endoderm; Arrow = Thyroid, Arrowhead = Lung. **d** Whole-mount confocal micrographs of mouse foregut explants. Foreguts were explanted at the 6–8 somite stage and cultured for 3 days. E-cad co-staining marks foregut endoderm; Arrow = Thyroid, Arrowhead = Lung.

specification, whereas the PI3-kinase (PI3K) pathway was upregulated in specified thyroid progenitors (Fig. 4a; Supplementary Fig. 3C). Upregulation of the Hippo pathway in the lung domain was unlikely to be an artifact of tissue digestion leading to release of junctional Yap/Taz reserves as this pathway was not upregulated in either thyroid epithelium or forebrain, tissues that underwent the same processing. Moreover, several pathways that were downregulated upon lung specification were related to lipid metabolism (Fig. 4a; Supplementary Fig. 3A).

To gain further insights into these findings, we compared the expression of select Hippo-, Tgf-β superfamily-, and Wnt-related genes in the lung primordium and foregut endoderm (Fig. 4b). Bmp receptor (*Bmpr1b*, *Bmpr2*) expression was upregulated in the lung primordium along with several genes involved in inhibition of Tgf-β signaling or bioavailability (*Dcn*, *Thbs1*, *Tgfbi*, *Ltbp1*) raising the possibility that both Bmp signaling activation and Tgf-β signaling inhibition are involved in lung specification. The upregulation of several direct Wnt targets (*Nkd2*, *Vcan*, *Nrcam*, *Twist1*) and downregulation of Wnt inhibitors (*Dkk1/3* [45] and *Sfrp5*) point to activation of Wnt signaling within the lung primordium, but not the thyroid. Finally, Hedgehog signaling was also found to be differentially regulated (Supplementary Fig. 4A).

To functionally evaluate our bioinformatics findings, we combined mouse foregut explant culture with pharmacological or genetic manipulation of specific pathways (Fig. 4c, d). Previous work both in vivo and in vitro has shown that Wnt is required for the lung, but not thyroid specification, Bmp signaling is involved in specification of both endodermal domains[3–5,9,21], and FGF-dependent PI3K signaling is required for thyroid, but not lung specification[21]. We first used a Shh driver to delete β-catenin (Shh[Cre]: βcat[flox/flox]), thereby abolishing β-catenin-dependent Wnt signaling within the majority of the anterior foregut domain (Fig. 4c). As expected, this resulted in elimination of the Nkx2-1[+] lung epithelial domain, similar to prior findings using genetic Wnt loss-of- function mouse models[4,5]. Although the Nkx2-1[+] thyroid domain was not affected, based on prior work profiling Shh kinetics in thyroid organogenesis[46], it is likely that the Shh[Cre] driver is not expressed in the thyroid primordial zone at the stage of thyroid lineage specification (Fig. 4c). Therefore, we applied the well-documented Wnt inhibitor, XAV-939[47], to mouse foregut cultures at this developmental stage to test the effect of Wnt loss of function throughout the foregut and observed the same phenocopied effect of abrogated lung specification while retaining thyroid specification (Fig. 4c). Surprisingly, absence of Wnt signaling in vivo at the moment of lung specification caused expansion of the Pdx1[+] domain (Supplementary Fig. 4B). We then asked whether Shh signaling was involved in lung specification as implied by our bioinformatics analysis. Use of cyclopamine to block Hedgehog signaling at 6–8 somite stage indeed resulted in lung agenesis, without affecting thyroid specification (Fig. 4d). We were able to rescue lung specification in the presence of cyclopamine by addition of CHIR99021, a specific GSK-3β inhibitor that activates Wnt signaling by stabilizing β-catenin, indicating that Wnt acted downstream of

Shh signaling. These findings are in keeping with prior work that established that endodermally secreted Shh acts indirectly by inducing hedgehog signaling in adjacent foregut mesenchymal cells which then secrete canonical Wnt ligands, which in turn act on the foregut endoderm to promote lung epithelial lineage specification[48]. In contrast to a requirement for Wnt signaling, we found that addition of several FGFs (FGF2, FGF7, and FGF10) to cyclopamine-treated foregut explants did not rescue lung specification (Fig. 4d), validating our previous findings[9] that FGF signaling is neither necessary nor sufficient for lung lineage specification in vitro.

Overall, our pathway analysis showed direct activation of the Bmp and Wnt pathways within the specified mouse lung primordium, a finding that is in keeping with both in vivo loss-of-function studies[3–5] and in vitro minimal signaling requirements for PSC derivation of lung primordial progenitors[9]. As the Hippo pathway was found to be differentially regulated at this early stage of lung development, we performed YAP/TAZ staining in the foregut endoderm at around E9.0 (Supplementary Fig. 4C). YAP/TAZ was cytoplasmic in most NKX2-1[+] cells but nuclear in the adjacent foregut, raising the possibility that transient YAP/TAZ activation immediately precedes lung specification though previous studies have suggested Yap/Taz dispensability for lung specification[49]. Lastly, our pathway analysis supports the apparent differential requirement for PI3K signaling in determination of thyroid fate and its dispensability for lung fate[21].

## Similarity model reveals deficiencies in in vitro lung specification. 
Having defined the genetic program of in vivo mouse primordial lung progenitors and the signaling pathways that are involved in lung specification in vivo, we next revisited our published mouse lung-directed differentiation protocol to assess the resemblance of our PSC-derived lung progenitors to in vivo lung primordial progenitors[9].

To this end, we employed our own computational approach, namely Linear Algebra Projections (LAP), which we had successfully applied in our previous work[43]. As this method relies on global gene expression of in vitro populations and their comparison to a reference basis of in vivo cell types, we first updated the reference basis of 61 cell types used in Lang et al.[50] to include our E8.25 ectoderm, E8.25 foregut endoderm, E13.5 Nkx2-1+thyroid, and E9.0 Nkx2-1[+] lung and forebrain bulk RNA-Seq data as well as an E16.5 lung epithelial cell data set[51] (six additional data sets). We also included single-cell data from E9.0 cells from mixed lineages captured separately from dissected thyroid, lung, and forebrain domains (eight clusters, see the Methods section) as well as single-cell RNA-Seq data from flow cytometry-enriched lung Nkx2-1[GFP+] cells at E13.5 (five clusters), E15.5 (five clusters) and E17.5 (six clusters)[52] (Supplementary Fig. 6). Our analysis showed that the single-cell clusters from the E9.0 lung and thyroid domains were almost entirely Nkx2-1[−] as expected given the rarity of GFP+progenitors at this

developmental stage. Therefore, these comparators did not overlap with the Nkx2-1$^+$ populations studied by bulk RNA-Seq. Thus, we created an updated reference basis of 91 in vivo cell populations, including mouse embryonic and adult populations from diverse stages, domains, and cell types.

As detailed in the methods, LAP scores are robust to comparisons across sequencing platforms (i.e., RNA-Seq versus microarrays). To further assess the robustness of the LAP method, following the inclusion of additional cell types and single-cell data, we re-analyzed the data from our previous work that had shown mouse ESC-derived thyroid progenitors have strong similarity to E13.5 mouse thyrocytes[43] (Supplementary Fig. 7B). Day(D)14 thyroid progenitors projected strongly on E13.5 Nkx2-1$^+$ thyrocytes, whereas D1 cells (cells exiting pluripotency) had a high undifferentiated-ESC similarity score.

We then proceeded with LAP analysis of the genome-wide microarray data of Nkx2-1$^+$ cell population derived in our minimal lung specification media (Wnt3a/Bmp4/RA) on 2D gelatin (2D-Nkx2-1$^+$)[9] (Fig. 5a). This population expressed key genes of lung endoderm identity as previously published[9], but was surprisingly found to also have high projection scores for mesenchymal-like cell populations such as mouse embryonic fibroblasts (MEFs), and a low score when compared with the E9.0 Nkx2-1$^+$ lung primordium program, indicating that the lung specification stage might be suboptimal (Fig. 5a).

We thus sought to troubleshoot our protocol for differentiating PSCs into lung primordial progenitors, focusing specifically on approaches that might reduce the aberrant mesenchymal program present in our in vitro engineered cells. As the "ADHERENS JUNCTION" pathway was involved in in vivo lung specification (Fig. 4a) and adherens junctions formation has been implicated in PSC cell-fate decisions[53,54], we hypothesized that the fidelity of in vitro-derived lung progenitors could be improved by appropriate modulation of the substratum biomechanical properties, which could minimize the mesenchymal program and preserve an Epcam+epithelial program. We further hypothesized this process might depend in part on the main epithelial adherens junction cadherin, CDH1.

To this end, we used the Nkx2-1$^{mCherry}$ mouse embryonic stem cell (ESC) line to specify Nkx2-1$^+$ lung progenitors in Wnt3a/Bmp4/retinoic acid (RA) containing media on either gelatin-coated ("2D Gelatin" condition) or thick Matrigel-coated ("3D Matrigel" condition) plates (Fig. 5b). We noticed higher frequency of epithelial-like Nkx2-1$^{mCherry+}$ colonies in the 3D vs. the 2D condition as early as D8 of the protocol (D2 of the specification stage) and this pattern was maintained by D14, which is the time point for lung primordial progenitor purification (Fig. 5c, d). Lung specification under 3D conditions brought about an increase in frequency of Nkx2-1$^{mCherry+}$Epcam$^+$ epithelial lung progenitors compared with 2D conditions and increased the yield of such progenitors per input undifferentiated (D0) ESC (Fig. 5e, f). When "3D Matrigel" Nkx2-1$^{mCherry+}$Epcam$^+$ cells were sorted to purity on D13-14 and embedded in Matrigel in the presence of lung differentiation media, they maintained high Epcam expression and gave rise to a variety of respiratory lineages, as shown by high expression levels of proximal (Scgb3a2, Scgb1a1, p63) and distal (Sftpc, Sftpb) lung markers and absence of thyroid markers (Pax8) (Fig. 5g). To investigate the implication of adherens junctions in the observed phenotype (Fig. 5d), we specified lung progenitors on three different substrata (2D-Gelatin, 2D-Matrigel (thin Matrigel layer), and 3D-Matrigel), and co-stained for the main epithelial adherens junction cadherin, E-cadherin (CDH1), and NKX2-1 (Supplementary Fig. 5A). E-cadherin$^+$ clusters contained the majority of NKX2-1$^+$ cells and their highest frequency was observed in the "3D Matrigel" conditions, indicating that both Matrigel and a low elastic modulus were probably required for the

increased derivation of Nkx2-1$^{mCherry+}$Epcam$^+$ lung progenitors. When adherens junction formation was inhibited by repeated addition of an anti-CDH1 monoclonal antibody during lung specification on "3D Matrigel", there was concomitant decrease in the Nkx2-1$^{mCherry+}$EPCAM$^+$ percentage, implying that adherens junction formation was necessary for the specification of the putative lung primordial epithelial population (Supplementary Fig. 5B, C).

Taken collectively, these data demonstrate that in vitro application of information gleaned from the in vivo characterization of lung primordial progenitors leads to identification of deficiencies in the derivation of lung multipotent epithelial progenitors from PSCs.

**Fidelity of lung progenitors derived under improved conditions**. We then asked whether the increased epithelialization of lung progenitors during in vitro lung specification resulted in an improved lung primordial progenitor signature.

To generate the in vitro transcriptome data required for LAP analysis, we performed bulk RNA-Seq of all epithelial cell populations from both 2D-Gelatin and 3D-Matrigel conditions, namely 2D-Nkx2-1$^+$EPCAM$^+$ (lung epithelial progenitors derived on 2D gelatin), 3D-Nkx2-1$^+$EPCAM$^+$ (lung epithelial progenitors derived on 3D Matrigel), and 3D-Nkx2-1$^-$EPCAM$^+$ (non-lung epithelial progenitors derived on 3D Matrigel) (Fig. 5h, i). The three populations were shown to be quite distinct both by hierarchical clustering (Fig. 5h) and principal component analysis (PCA) (Fig. 5i). Interestingly, the 3D-Nkx2-1$^-$EPCAM$^+$ (non-lung epithelium) was clearly separated from the two Nkx2-1$^+$ populations along the PC1 axis (82% variance) in the PCA plot. For direct comparison between the old and new lung specification protocols (Fig. 6a), we included again the 2D-Nkx2-1$^+$ condition in the assessed conditions[9].

Enrichment of epithelial progenitors in our lung differentiation protocol correlated with improved primordial progenitor fidelity as reflected by increased similarity scores to the E9.0 in vivo Nkx2-1$^+$ lung primordium of all EPCAM$^+$ populations (Fig. 6b). Nevertheless, the 3D-Nkx2-1$^-$EPCAM$^+$ population had lower similarity scores to other lung epithelial cell populations and higher foregut endoderm and liver similarity scores compared with the Nkx2-1$^+$EPCAM$^+$ populations (Fig. 6b, c). Notably, the mesenchymal (MEF) signature that was prominent in Nkx2-1$^+$ lung progenitors derived under non-optimal biomechanical conditions all but disappeared in EPCAM$^+$ cells under improved culture conditions (Fig. 6d; Supplementary Fig. 6A, B). None of the four tested populations showed high similarity to neural or thyroid cell populations (Supplementary Fig. 7A), indicating that our Wnt3a/Bmp4/RA specification media does indeed produce lung-fated cells.

The two Nkx2-1$^+$EPCAM$^+$ populations also had high projections on distinct lung embryonic epithelial populations, with 2D-Nkx2-1$^+$EPCAM$^+$ having a higher similarity score to a proximal E13.5 subpopulation (enriched in Scgb3a2, Sox2, Sema3c, and Cldn10) and 3D-Nkx2-1$^+$EPCAM$^+$ projecting more strongly on an E17.5 distal lung epithelial subpopulation (Supplementary Fig. 6A, B; Fig. 6b). This suggests the culture conditions of in vitro lung epithelial primordial progenitors may also affect downstream cell-fate decisions.

To gain further confidence in the similarity scores described above, we applied an independent orthogonal signature projection method based on gene set analysis using the bulk RNA-Seq or microarray data sets in Fig. 6b–d and Supplementary Fig. 7A. As the overall patterns of radar plots in Supplementary Fig. 7C demonstrate, this method was able to replicate the LAP analysis for our control cell populations (D1,

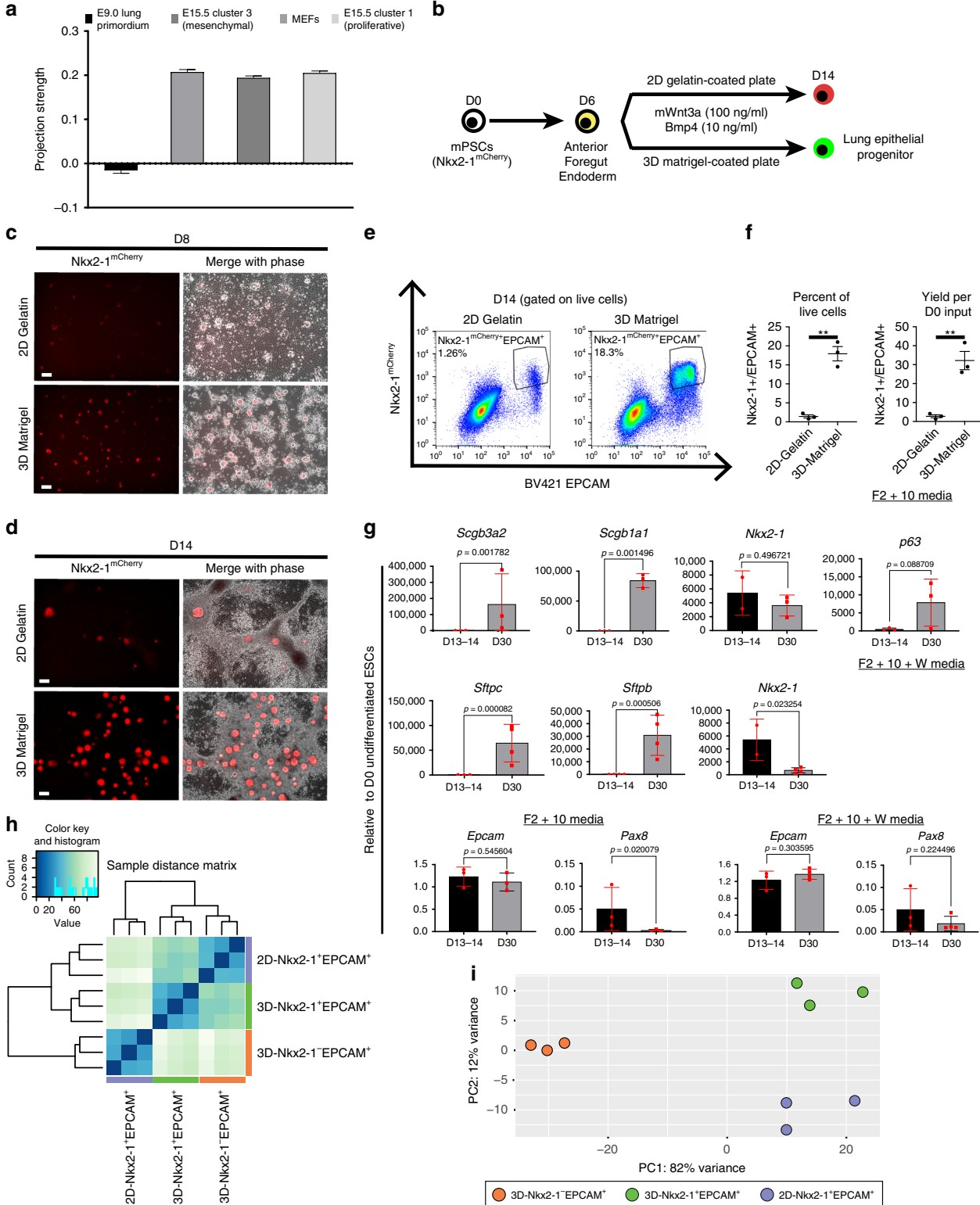

D7, and D14 from Dame et al.[43]) with high projections on undifferentiated ESC and E8.25 ectoderm for D1 cells and high E13.5 thyroid projection for D14 cells. Similarly, a reduction of the MEF score and increases in E9.0 Lung Primordium and E8.25 Foregut Endoderm scores was observed in all EPCAM+ populations whereas the 3D-Nkx2-1-EPCAM+ population also showed an increased liver score (Fig. 6e).

Thus, our unbiased, genome-wide similarity analysis allowed quantitative comparisons of PSC-derived lung and thyroid engineered progenitors to reference benchmarks including rare primordial Nkx2-1+ in vivo progenitors and demonstrated derivation of epithelial progenitors with bona fide lung primordial progenitor identity in our modified lung-directed differentiation protocol.

**Fig. 5 ECM modulation during lung specification. a** Projection strength scores for the PSC-derived lung progenitors described in Serra et al.[9]. **b** Schematic illustrating directed differentiation of mouse PSCs into lung epithelial lineages using 2D-gelatin-coated plates as well as 3D-Matrigel-coated plates. For these experiments, we used a mouse ESC line with an Nkx2-1[mCherry] reporter. The 2D condition leads to a low yield of Nkx2-1[mCherry+]/Epcam[+] cells and a dominant population of Nkx2-1[mCherry+]/Epcam[−] cells. **c, d** 3D Matrigel leads to increased numbers of Nkx2-1[mCherry+] cells compared with 2D gelatin at both D8 and D14, Scale bar: 100 μm. **e** Bivariate flow cytometry plots for D14 lung progenitors showing expression of Nkx2-1[mCherry] and BV421-Epcam. **f** Quantification of Nkx2-1[mCherry+]/Epcam[+] populations shown in **e** as a percent of cells (left panel) and cell yield per starting (D0) undifferentiated ESC (right panel), P-value < 0.01 by two-tailed t tests, N = 3 independent experiments, error bars represent standard deviation. **g** RT-qPCR data from downstream differentiation of 3D-Matrigel Nkx2-1[mCherry+]Epcam[+] sorted populations shown in **d**. Fold changes relative to undifferentiated cells, statistics from two-tailed, unpaired t tests, N = 3 independent experiments, error bars represent standard deviation. **h** D14 EPCAM[+] cells from both 2D-gelatin-coated and 3D-Matrigel-coated plates were assessed by bulk RNA-seq. The resulting samples were then hierarchically clustered using Euclidean distances. **i** Principal Component Analysis (PCA) of bulk RNA-Seq samples. The top 500 most variable genes were used in this analysis and they are listed in Supplementary Data 2.

## Discussion

Directed differentiation of PSCs has emerged as one of the most promising regenerative medicine platforms[55]. One of the remaining obstacles in the clinical use of PSCs for the treatment of lung disease is the partial understanding of the sequence of cell-fate decisions that lead from PSC-derived anterior foregut endoderm to functional, clinically relevant lung cell populations. In vivo, the formation of all lung epithelial lineages depends on the critical moment of lung specification. An incomplete understanding of this particular progenitor population can limit attempts to generate a similar population in vitro. By providing global transcriptomic signatures of the earliest in vivo Nkx2-1[+] progenitors and quantitative algorithms for their application, our work establishes important benchmarks against which cells engineered in vitro can be compared as they proceed through the developmental gateway of lineage specification. Consequently, our current work addresses deficiencies in the development of lung differentiation protocols by applying computational methods to improve the efficiency of in vitro lung progenitor derivation.

The underlying assumption of current lung-directed differentiation protocols is that lung epithelial lineages in vivo are derived through an Nkx2-1[+] intermediate primordial progenitor[6–8,20,56]. We substantiated this statement by performing indelible marking of Nkx2-1[+] lung progenitors around E9.0 using an inducible Nkx2-1 knock-in Cre driver and an nT/nG reporter mouse. Nuclear GFP-positive cells were exclusively found in the epithelium (EPCAM[+]) of the developing lung at E14.5 and in E18.5 cells expressing markers of lung epithelial lineages. These findings further support the wide use of Nkx2-1 as the earliest marker of lung epithelial fate and strongly imply that the use of Nkx2-1 fluorescent reporters in mouse and human PSCs[7,20] in combination with lung specification cocktails leads to derivation and isolation of putative lung progenitors with broad differentiation competence.

Using our Nkx2-1[GFP] knock-in mouse, we sorted Nkx2-1[+] lung epithelial progenitors at E9.0 and characterized their transcriptome by bulk RNA-Seq. The simultaneous purification and characterization of other Nkx2-1[+] embryonic progenitors and E8.25 precursor populations (foregut endoderm and ectoderm) allowed us to define the genetic program of primordial lung progenitors and signaling pathways that appear to be peculiar to lung specification, relative to early endoderm and thyroid epithelium. Previous work in *Xenopus* and mouse has elucidated signals, such as RA, that impart lung competence to foregut endoderm[48] as well as lung specification signals emanating from the mesenchyme, such as Bmp and Wnt[3–5,9]. Our pathway analysis does demonstrate activation of Wnt and Tgf-β superfamily pathways in lung primordial progenitors versus foregut endoderm vindicating the use of Wnt and Bmp activators in in vitro lung specification cocktails for PSC-based systems.

Although the Shh pathway is also upregulated in primordial lung progenitors, our ex vivo foregut explant experiments demonstrate that it acts upstream of β-catenin-dependent Wnt signaling and may be redundant for in vitro PSC-derived foregut specification as long as Wnt activators are present. Interestingly, retinol metabolism is downregulated in lung progenitors which may imply that RA signaling is necessary for making foregut cells competent to adopt lung fate, but it is not required during lung specification as reported previously[48]. As far as pathways regulating thyroid versus lung endodermal fates are concerned, the PI3K-mediated signaling features prominently in thyroid-related (3/10 pathways), but is absent from lung specification-related pathways. This is experimentally supported by the inability of FGF ligands to rescue lung ablation following inhibition of Shh signaling in foregut explants as well as previous work from our group[9,21]. More specifically, inhibition of PI3K-dependent FGF signaling in either *Xenopus* or in mouse ESC-derived anterior foregut at the moment of thyroid specification completely abrogated thyroid fate[21].

Intriguingly, the Hippo pathway is also differentially expressed during lung specification and various YAP/TAZ targets such as *Ankrd1* and *Cyr61* are upregulated, whereas YAP/TAZ negative regulators are downregulated in lung primordial progenitors relative to pre-specified foregut endoderm. Yet, the significance of these findings is, at present, unclear as Shh[Cre]-driven *Yap* deletion within the foregut endoderm does not lead to lung agenesis[49].

Although pathway analyses of in vivo lung progenitors reveal that the minimal signals necessary and sufficient to induce lung fate in foregut progenitors have already been present in our lung specification media, the mesenchymal-like phenotype of in vitro-derived mouse lung progenitors implied that additional signals were required for proper preservation of the epithelial program during lung specification. The importance of biomechanical cues in cell-fate decisions of embryonic progenitors is increasingly recognized with a recent example of the role of the extracellular matrix in endocrine versus ductal fate decisions of bipotent pancreatic embryonic progenitors[57]. In this work, modulation of both the composition and the elastic modulus of the substratum resulted in increased derivation of EPCAM[+] lung epithelial progenitors with broad differentiation potential. Future studies are required to decipher which of these variables is responsible for the improved yield of lung epithelial progenitors and in turn the signaling mechanisms regulated by these variables. Furthermore, it is still unclear whether this yield results from selection of an epithelial progenitor subset or inhibition of epithelial-to-mesenchymal drift in all cells.

Having transcriptionally defined E9.0 in vivo lung epithelial primordial progenitors as a benchmark for lung primordial progenitors, this work attempted then to define the fidelity of in vitro lung progenitors derived under improved biomechanical conditions. As the repertoire of in vitro engineered cell types

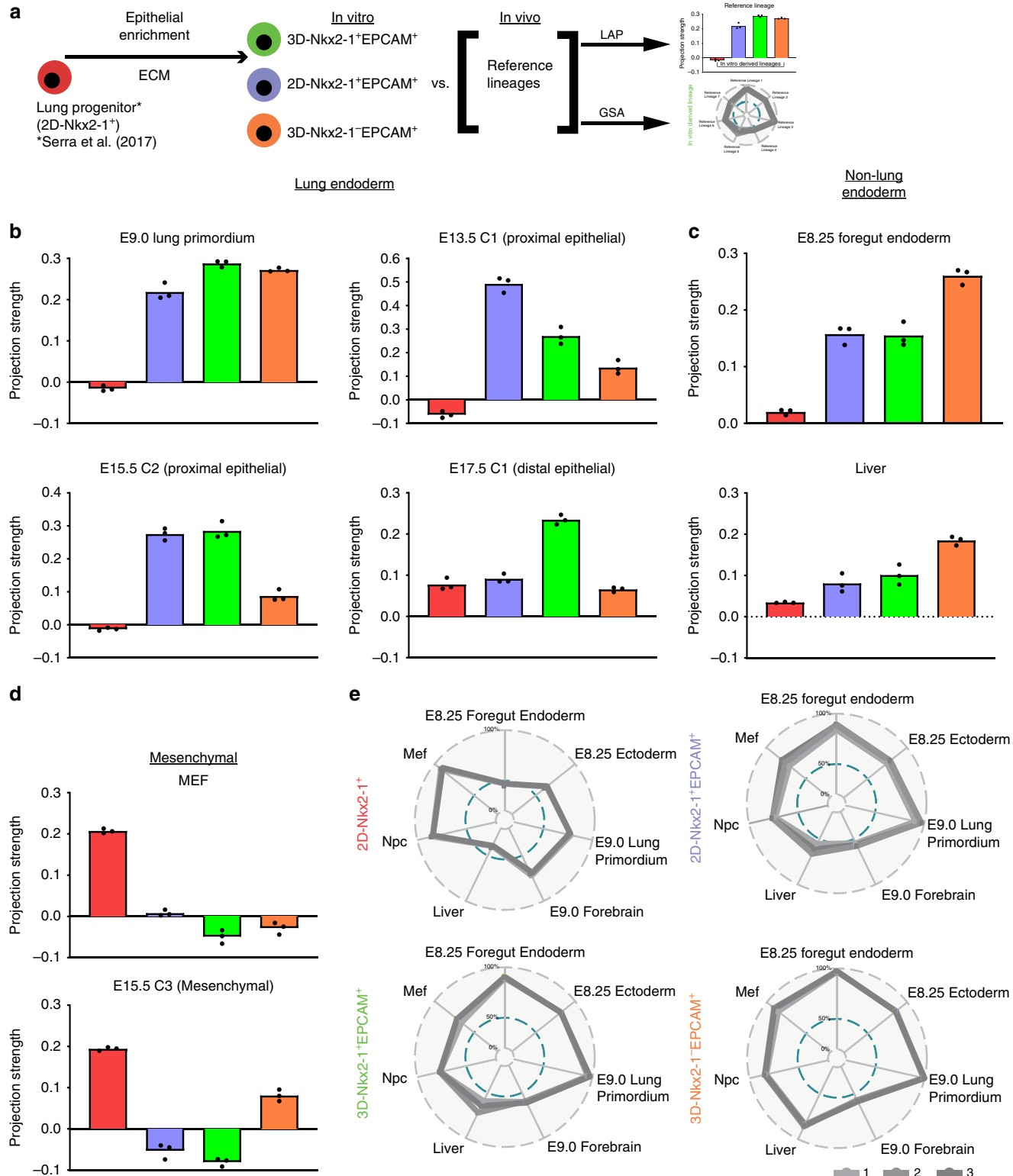

**Fig. 6 Similarity analysis of in vitro PSC-derived and in vivo lung progenitors. a** Schematic depicting the four types of in vitro PSC-derived progenitors using a lung specification cocktail (in vitro-derived lineages) and derivation of similarity scores to in vivo populations (reference lineages) via two methods, namely linear projection analysis (LAP) and gene set analysis (GSA) (**b–d**). Projections strength scores from LAP analysis for the four populations shown in **a**, including projections to lung endodermal populations (**b**), non-lung endoderm (**c**), and mesenchymal-like populations (**d**). Scores shown in Fig. 5a are re-plotted here for comparison purposes. **e** Projection scores from gene set analysis using the reference cell populations from **b–d** and Supplementary Fig. 7A with bulk RNA-Seq data.

through directed differentiation, direct conversion, and forward programming is rapidly expanding, it is essential to define the correspondence of in vitro-derived and in vivo cell types. Several methods, including our own work, have been developed that apply network analysis to cell-fate conversions[58], identify TFs that underlie cell identity[59,60], and predict reprogramming TFs[50,61]. Previously, we successfully used LAP to interrogate the degree to which thyroid progenitors derived from PSCs were similar to E13.5 thyrocytes[43]. Application of the same method in the characterization of in vitro lung progenitors showed that enhancement of their epithelial properties resulted in increased resemblance to in vivo lung primordial progenitors and highly reduced mesenchymal cell signature. Nkx2-1+EPCAM+ progenitors derived under 2D and 3D conditions had similar scores that were greatly improved compared to Nkx2-1+ progenitors without EPCAM enrichment. This implies that most of the lung competence in the 2D, suboptimal condition appears to be in the small (~1–2%) Nkx2-1+EPCAM+population. Nevertheless, differential similarity scores to epithelial lung progenitors from later time points indicate that culture on different substrata may introduce fate biases, a tantalizing possibility that merits further study. The increased similarity to in vivo lung primordium of even the Nkx2-1−EPCAM+ population may seem perplexing at first glance. Yet, the fact that this population does not project on other lung epithelial lung populations, and has a higher liver signature suggests that Nkx2-1-negative cells are qualitatively different than Nkx2-1+ epithelial progenitors at the same stage of in vitro directed differentiation. It is also possible that the sorted Nkx2-1−EPCAM+ population contains less differentiated cells that are about to adopt a lung fate, as suggested by the higher foregut projection of this population.

In summary, this work defines the genetic program of in vivo primordial lung progenitors and uses similarity analysis to increase the correspondence of PSC-derived lung progenitors with their in vivo counterparts via manipulation of their biomechanical microenvironment. Our findings and the proposed methodology provide a framework for rational and systematic development and refinement of PSC-directed differentiation protocols and can propel future studies of lung primordial progenitors.

## Methods

**Animal maintenance**. All mouse studies involving mice carrying GFP, CreERT2, Cre, and nT/nG transgenes were approved by the Institutional Animal Care and Use Committee of Boston University School of Medicine. All mouse studies involving mouse foregut explants were approved by the Institutional Animal Care and Use Committee of Cincinnati Children's Hospital. We previously published the generation of Nkx2-1GFP mice (maintained on a C57BL/6J background)[7]. The Nkx2-1CreERT2 (Nkx2-1tm1.1(cre/ERT2)Zjh/)[27], R26RnT/nG (B6;129S6-Gt(ROSA) 26Sortm1(CAG-tdTomato*,-EGFP*)Ees/)[28], Nkx2.1-Cre (C57BL/6J-Tg (Nkx2-1-cre) 2Sand/J)[26], and R26R (B6.129S4-Gt(ROSA)26Sortm1Sor/J)[62] mouse strains were purchased from Jackson Laboratory.

**Characterization of the Nkx2-1GFP mouse**. To characterize the expression of Nkx2-1GFP during mouse lung development, timed-pregnant females (2–6 months old) were euthanized by inhalation of isoflurane (purchased from IsoThesia, Henry Schein, Dublin, OH) and subsequent cervical dislocation between embryonic (E) day 8.5 and E18.5, and embryos were delivered by cesarean using Dumont dissection forceps (Fine Science Tools, Foster City, CA). The lung domain was dissected out using Tungsten needles (Fine Science Tools) under an Olympus stereo fluorescence imaging microscope (model SZX16, Olympus America, Central Valley, PA) and fluorescence, dark- and bright-field micrographs were captured using the QCapture software (QImaging, Surrey, BC, Canada) and saved as TIFF files. Post-acquisition image processing, including image merging, was done using the SPOT Advanced Software (SPOT Imaging Solutions, Sterling Heights, MI).

Adult male or female Nkx2-1GFP mice (up to 12 months old) were characterized by immunohistochemistry and flow cytometry. Adult mice were euthanized and the lungs were inflation-fixed with 4% paraformaldehyde and extracted. The following day, they were embedded in Optimum Cutting Temperature (OCT) embedding medium to preserve GFP fluorescence and snap-frozen. In all, 6-μm cryosections were prepared using a cryostat. For staining, cryosections were rinsed

with PBS, permeabilized with 0.3% Triton-X and stained overnight at 4 °C with the following antibodies: goat anti-SCGB1A1 (1:200, Santa Cruz, Cat no.: sc-9772), rabbit anti-pro−SPC (1:200, Seven Hills Bioreagents, Cat no.: WRAB-9337), hamster anti-mouse PDPN1 monoclonal antibody (1:200, Developmental Studies Hybridoma Bank, 8.1.1), rabbit anti-acetylated α-tubulin (1:800, Cell Signaling Technology, 5335), rabbit anti-P63-alpha (1:100, Cell Signaling Technology, 13109), rabbit anti-NKX2-1 (1:200, Abcam, Cat no.: ab76013), and for certain sections a chicken anti-GFP (1:50, Invitrogen, A10262). The following day, sections were incubated with either Alexa Fluor secondary antibodies (donkey anti-goat or anti-rabbit for SCGB1A1, pro-SPC, P63, NKX2-1, and acetylated α-tubulin antibodies; 1:200, Molecular Probes, Invitrogen), or with an anti-hamster biotinylated antibody followed by incubation with an anti-biotin Cy3 antibody for the anti-PDPN antibody, counterstained with DAPI and imaged on a confocal Zeiss LSM 510 Meta NLO laser scanning inverted microscope using the LSM510 software or on a Zeiss LSM 700 Laser Scanning Confocal Microscope using the ZEN software (Carl Zeiss Microscopy, Thornwood, NY). Post-acquisition image processing was performed in ZEN 2.1 (black edition) software.

Lungs from adult Nkx2-1GFP mice were extracted, minced, and digested using Collagenase A (0.1%, Roche 103578) and Dispase II (2.4 U ml−1, Roche 295825) supplemented with 2.5 mM CaCl2 for 60 min at 37 °C with periodic trituration. Monodispersed cells were resuspended in HBSS with 2% FBS (HBSS + buffer), filtered through 30-μm FACS strainers (Miltenyi Biotech, Cat no.: 130-041−407) and stained with an anti-EPCAM antibody (APC/Cy7 anti-mouse CD326, clone G8.8, Biolegend, Cat no.: 118218, 0.2 mg ml−1, 1:80) or an isotype control (APC/Cy7 Rat IgG2a, κ Isotype, clone RTK2758, Cat no.: 400524, 0.2 mg ml−1, 1:80) for 30 min on ice. Sorting was performed at the Boston University Flow Cytometry Core Facility using the MoFlo cell sorter and cells were stored at −80 °C until further analysis.

**Real-time quantitative polymerase chain reaction**. RNA from sorted cells was extracted using QIAshredder columns (QIAGEN, Mansfield, MA, Cat no.: 79656) and the RNeasy Plus Mini Kit (QIAGEN, Cat no.: 74136) according to the manufacturer's instructions. Purified RNA was eluted in either 14 or 30 μl nuclease-free water, quantified on a NanoDrop ND-1000 microvolume spectrophotometer (ThermoFisher Scientific, Waltham, MA), and diluted as needed. cDNA was prepared using Taqman Reverse Transcription Reagents (Applied Biosystems, Waltham, MA, Cat no.: N808-0234). cDNA samples were prepared with TaqMan Fast Universal Master Mix (Life Technologies, 4367846) and diluted to 6.25 ng per 25 μl reaction for real-time analysis on the StepOnePlus Real Time PCR System (Applied Biosystems). Relative expression analysis calculations were performed according to the ΔΔCt method[63] utilizing 18S rRNA as the internal reference gene and either presort cells or whole embryo cells as the reference sample. If expression was undetected based on the set threshold, the value was set to the maximum number of cycles (40) to allow fold change calculations. The data were graphed for visualization as shown in the figures using the GraphPad Prism 8 (version 8.0.2) software.

TaqMan gene expression arrays were purchased from Applied Biosystems and are listed in Supplementary Table 1.

**Statistical analysis**. Error bars in graphs represent SD or SEM as indicated in the figure legends. Biological sample replicates ($N$) are also indicated in the legends. Statistically significant differences between conditions (dCt values used for RT-qPCR calculations) were determined using two-tailed unpaired Student's $t$ tests or as specified in figure legends. Statistical significance is represented as $P$-values.

**Lineage trace**. To trace the contribution of lung primordial progenitor cells to lung epithelial cell lineages, Nkx2-1CreERT2 mice were crossed with R26RnT/nG mice. For these experiments, we used 5–8-month-old male mice and 3–6-month-old female mice. Tamoxifen (Sigma-Aldrich, St. Louis, MO, Cat no.: T5648) was dissolved in ethanol at room temperature, and then diluted 9:1 with corn oil to a final concentration of 10 mg ml−1 tamoxifen. Recombination was induced by two doses of tamoxifen (2.5 mg tamoxifen per 10 g of body weight) delivered by intraperitoneal injection at E7.5 and E8.0, and embryos were harvested at either E13.5 or E14.5. Lungs were either imaged using an Olympus stereo fluorescence imaging microscope or cryo-embedded and sectioned. Cryosections were stained with an APC-Cy7-conjugated anti-mouse rat EPCAM antibody (Biolegend CD326, Cat no.: 118217) and imaged using a Nikon Eclipse Ni upright microscope and the NIS-Elements software (Nikon Instruments Inc., Melville, NY).

To counteract estrogenic effects of tamoxifen and prevent dystocia in pregnant animals at later time points, we modified our protocol by adding progesterone and switching the administration route to oral gavage. Tamoxifen and progesterone were dissolved in ethanol at room temperature, and then diluted 9:1 with corn oil to a final concentration of 20 mg ml−1 tamoxifen with 10 mg ml−1 progesterone. Recombination was induced by a single dose of tamoxifen–progesterone (2 mg tamoxifen and 1 mg progesterone per 10 g of body weight) delivered by oral gavage to pregnant females at E8.75. Control animals received either a single dose of tamoxifen–progesterone (2 mg tamoxifen and 1 mg progesterone per 10 g of body weight) at E7.5 or a gavage of 9:1 corn oil and ethanol (100 μl per 10 g of body weight) at E8.75.

Embryos receiving tamoxifen–progesterone or corn oil at E8.75 were harvested at E18.5, and lungs, brain, stomach, esophagus, and liver were dissected and evaluated for GFP expression by fluorescent microscopy. Embryos receiving tamoxifen–progesterone at E7.5 were harvested at E13.5 and sectioned intact to evaluate GFP expression in the lungs and throughout the embryo. Tail tissue was used to confirm the genotype of each embryo. Dissected organs or whole embryos were fixed overnight in 4% paraformaldehyde at 4 °C, rinsed in PBS, transferred to 30% sucrose, cryo-embedded in OCT medium, and sectioned at 5 μm thickness. For immunohistochemistry, sections were rehydrated and then rinsed, permeabilized with 0.3% Triton-X, rinsed again, and incubated overnight at 4 °C with the following primary antibodies: rabbit anti-pro-SPC (1:1000, Seven Hills Bioreagents Cat no.: WRAB-9337), Syrian hamster anti-PDPN (1:200, eBiosciences, clone 8.1.1), goat anti-SCGB1A1 (1:200, Sigma-Aldrich, Cat no.: 07-623), rabbit anti-acetylated α-tubulin (1:800, Cell Signaling Technology, Cat no.: 5335), rabbit anti-P63-alpha (1:100, Cell Signaling Technology, Cat no.: 13109), anti-E-Cadherin (phospho S838+S840) (1:200, abcam, Cat no.: ab76319), and rabbit anti-NKX2-1 (1:200, abcam, Cat no.: ab76013). Following incubation, sections were rinsed and staining was detected using fluorophore-conjugated (Alexa Fluor 647) secondary antibodies. All sections were counterstained with DAPI as a nuclear marker. Images were captured at either ×20 or ×40 magnification using a Zeiss LSM 700 Laser Scanning Confocal Microscope and ZEN software or Nikon Eclipse Ni upright microscope and the NIS-Elements software. Post-acquisition image processing was performed in the ZEN 2.1 (black edition) software or NIS-Elements Viewer 4.20.

**Isolation of embryonic progenitors for bulk RNA-Seq.** All embryonic progenitor populations were isolated from mouse embryos using the Nkx2-1[GFP] reporter mouse[7]. Briefly, breeding cages were set up with one male (up to 12 months old) and three female mice (2–6 months old), and the morning of the day a vaginal plug was detected was defined as embryonic day E0.5. Foregut endoderm and ectoderm cells were isolated from E8.25 embryos, forebrain and lung cells were isolated from E9.0 embryos and thyroid cells were isolated from E13.5 embryos (Fig. 2b). Embryo extraction was performed as described above.

For E8.25 embryos (5–6 somite stage), we used a published protocol to sort foregut endoderm and ectoderm[35]. Briefly, embryos were incubated in trypsin for up to 3 min, and the cell monodispersion was incubated with ENDM1 primary antibody (DMBC2-8-610, 1 mg ml−1, 1:100) and Cy3-conjugated secondary antibody (Cy3-AffiniPure Goat Anti-Rat IgG (H+L) Jackson Immunoresearch, Cat no.: 112-165-062, 1:100) for 30 min on ice each. The isotype control was a rat IgG1, kappa monoclonal antibody ([RTK2071], Abcam, Cat no.: ab18412, 0.5 mg ml−1, 1:50). Following a blocking step for 10 min at room temperature (ChromPure Rat IgG, whole molecule, Jackson Immunoresearch, Cat no.: 012-000-003, 1:100), cells were stained with an anti-EPCAM antibody (APC/Cy7 anti-mouse CD326, clone G8.8, Biolegend, Cat no.: 118218, 0.2 mg ml−1, 1:80) or an isotype control (APC/Cy7 Rat IgG2a, κ Isotype, clone RTK2758, Cat no.: 400524, 0.2 mg ml−1, 1:80) for 30 min on ice. Sorting was performed at the Boston University Flow Cytometry Core Facility using the MoFlo cell sorter and cells were stored at −80 °C until further analysis.

For E9.0 and E13.5 embryos, the three Nkx2-1[GFP]-expressing domains (forebrain, thyroid, and lung) were dissected out using Tungsten needles with the help of an Olympus stereo fluorescence imaging microscope and kept in HBSS with 10% FBS. Tissues were digested using Collagenase A (0.1%, Roche 103578) and Dispase II (2.4 U ml−1, Roche 295825) supplemented with 2.5 mM CaCl₂ for 60 min at 37 °C with periodic trituration. Monodispersed cells were resuspended in HBSS + buffer, filtered through 30-μm FACS strainers (Miltenyi Biotech, Cat no.: 130-041-407), and stained with propidium iodide for dead cell exclusion (1:500) directly before sorting. Sorting was performed at the Boston University Flow Cytometry Core Facility using the MoFlo or FACSARIA II SORP high speed cell sorters, and cells were stored at −80 °C until further analysis. Flow cytometry plots were generated using the FlowJo software (V10, Becton, Dickinson & Company).

**RNA-Seq preparation and analysis.** For each replicate (3 replicates per population) to be used in RNA-Seq, samples from several sorts were pooled. Sequencing libraries were prepared from the total RNA samples using Illumina® TruSeq® RNA Sample Preparation Kit v2. Briefly, the mRNA was isolated using magnetic beads-based poly(A) selection, fragmented, and randomly primed for reverse transcription, followed by second-strand synthesis to create double-stranded cDNA fragments. These cDNA fragments were then end-repaired, added with a single "A" base, and ligated to Illumina® Paired-End sequencing adapters. The products were purified and PCR-amplified to create the final cDNA library. The libraries from individual samples were pooled in groups of four for cluster generation on the Illumina® cBot using Illumina® TruSeq® Paired-End Cluster Kit. Each sample was sequenced four per lane on the Illumina® HiSeq 2500 to generate more than 25 million Paired-End 100-bp reads.

Reads were aligned to the mouse genome (GRCm38) and quantified using Subread[64]. Differences in library size were normalized using the Trimmed Mean of M−values method[65] from the edgeR BioConductor package[66]. Limma voom was used to perform differential expression analysis, using false discovery rate (FDR) < 0.05 as threshold for statistical significance. Pairwise comparisons are presented in Supplementary Data 1. Gene set enrichment analysis (GSEA) was performed on

KEGG pathways to assess the functional enrichment of several pairwise comparisons[67]. The list of KEGG pathways was curated to exclude disease-related, immune-related, and several metabolic pathways and the final list used in GSEA can be found in Supplementary Table 2.

Heatmaps of selected genes for specific pathways were created using the matrix visualization and analysis software, Morpheus (https://software.broadinstitute.org/morpheus/) and were exported as SVG files for further processing.

**Mouse lung-directed differentiation.** As previously published, Nkx2-1[mCherry] mouse ESCs[68] were differentiated into definitive endoderm over the course of 5 days in serum-free cSFDM medium with 50 ng ml−1 Activin A (R&D Systems, Cat no.: 338-AC) added from days 2.5 to 5[7,9,21]. The resulting embryoid bodies were plated onto P100 petri dishes in cSFDM supplemented with 100 ng ml−1 rmNoggin (R&D Systems, Cat no.: 1967-NG), and 10 μM SB431542 (Sigma-Aldrich, Cat no.: S4317) as previously described[7,9]. For Nkx2-1 induction, embryoid bodies were plated on six-well plates coated in gelatin, as previously described[9], or 300 μl of Matrigel (Corning 356231) at 10⁶ cells/well in cSFDM with 10 ng ml−1 rhBMP4 (R&D Systems, Cat no.: 314-BP), 100 ng ml−1 rmWnt3a (R&D Systems, Cat no.: 1324-WN), and 10 μM ROCK inhibitor (Tocris, Cat no.: Y-27632). Cells were then fed every other day with the same media minus the ROCK inhibitor. On D14, Matrigel was disrupted using 2 mg ml−1 dispase for 1 h, then spun down at 100 × g for 1 min to enrich for clusters of epithelial cells. These cells were then broken down into a single-cell suspension using 0.25% trypsin and stained with an anti-EPCAM antibody (Biolegend, Cat no.: 118225, 1:500) and the live/dead stain DRAQ7 (Biolegend, Cat no.: 424001, 1:100). EPCAM+/Nkx2-1[mCherry+] live cells were then enriched by fluorescence-activated cell sorting. For further differentiation, these cells were replated as a 3D suspension in Matrigel (Corning 356231) at 500 cells μl−1 Matrigel with cSFDM supplemented with 250 ng ml−1 rhFGF2 (R&D Systems, Cat no.: 233-FB), 100 ng ml−1 rhFGF10 (R&D Systems, Cat no.: 345-FG) and 100 ng ml−1 Heparin Salt (Millipore Sigma, Cat no.: H3393) (F2+10, proximalization media), or F2+10 media with 200 ng ml−1 rmWnt3a (R&D Systems, Cat no.: 1324-WN) (F2+10+W, distalization media), as previously described[9]. This media was supplemented with ROCK inhibitor for the first two days after replating, and then changed every 2–3 days thereafter. On D30, cells were imaged using a Keyence microscope, then harvested by disrupting Matrigel using 2 mg ml−1 dispase for 1 h, spinning down for 5 min at 300 × g, and then dispersed into a single-cell suspension using 0.25% trypsin. Harvested cells were stored at −80 °C until further analysis.

For experiments involving adherens junctions inhibition, a rat anti-CDH1 antibody (Sigma-Aldrich, Cat no.: U3254) was added every other day to the culture media from D6 to D14 at a concentration of 40 μg ml−1 [69]. Cells were harvested on D14 and processed for flow cytometry, as described above. For E-cadherin imaging, cells were cultured on either gelatin (2D Gelatin), a thin layer of Matrigel (2D Matrigel), or a thick layer of Matrigel (3D Matrigel) in two-well Nunc™ Lab-Tek™ II Chamber Slide™ (ThermoFisher Scientific, Cat no.: 154461PK) from D6 to D14. Cells were then fixed and stained with a rat anti-CDH1 antibody (Sigma-Aldrich, Cat no.: U3254) and a rabbit anti-NKX2-1 (1:200, abcam, Cat no.: ab76013) overnight at 4 °C. Staining was detected using fluorophore-conjugated (Alexa Fluor) anti-rat and anti-rabbit secondary antibodies. All wells were counterstained with DAPI as a nuclear marker. Tiled images were captured at ×10 magnification using a Zeiss LSM 700 Laser Scanning Confocal Microscope, and ZEN software and post-acquisition image processing was performed in ZEN 2.1 (black edition) software.

Flow cytometry plots were generated and formatted using the FlowJo software.

**Mouse foregut explants.** Whole foreguts were dissected from mouse embryos (6–8 somite pairs) in Hank's balanced salt solution (HBSS) then explanted onto 8-μm pore size Whatman Nucleopore Track-Etch Membranes (Millipore). Explants were cultured for 2–3 days in a base medium [BGJb medium (Gibco)+10% fetal bovine serum (FBS, Sigma) and 0.2 mg ml−1 ascorbic acid] containing either specific inhibitors or DMSO as a vehicle control. Whole-mount immunostaining was performed using a modification of the method of Ahnfelt-Ronne et al.[70]. The primary antibodies used were: guinea pig anti-Nkx2-1 (Seven Hill Bioreagents; 1:500) and rat anti-E-cad (R&D Systems; 1:2000). After staining, samples were cleared with Murray's clear (2:1 benzyl benzoate: benzyl alcohol) and imaged using a Nikon A1Rsi inverted laser confocal microscope. Imaris software was used to analyze the images.

**Single-cell RNA-Seq analysis.** In order to enrich the reference basis used for gene expression projections (see the next section), we leveraged two experiments previously performed in our labs. The E13.5, E15.5, and E17.5 single-cell data were generated as described in ref. [52]. Briefly, lungs from Nkx2-1[GFP] embryos of the respective embryonic day were harvested and processed into single-cell suspensions using an enzyme solution containing dispase (Collaborative Biosciences), collagenase I (Life Technologies), and RNAse-free DNAse (Promega). Nkx2-1[GFP]-positive cells stained with DAPI were sorted for GFP and also DAPI to exclude dead cells using a FACSJazz (BD Biosciences) flow cytometer using wild-type embryonic lung single-cell suspensions as a negative control. Cells were collected into FACS tubes containing cell growth media composed of 10% fetal bovine serum in DMEM/F12 media. Cells were loaded onto a GemCode instrument (10X

Genomics) to produce single-cell bar-coded droplets using ×10 Single Cell 3' v2 chemistry. Libraries generated were sequenced across at least one lane on the Illumina HiSeq 2500 instrument with the HiSeq Rapid SBS kit. The resulting reads were aligned and gene level unique molecular identifier (UMI) counts obtain using Cell Ranger version 2.0.1 (×10 Genomics).

For E9.0 single-cell RNA-Seq data, Nkx2-1$^{GFP}$-expressing domains (forebrain, thyroid, and lung tissues containing both GFP+ and GFP− lineages) were isolated from E9.0 embryos and monodispersed using the collagenase-dispase method, as described above. Cell suspensions were filtered through 30 µm FACS strainers (Miltenyi Biotech, Cat no.: 130-041-407) and stained with Calcein Blue (1:500) directly before sorting viable cells. Given the rarity of GFP+cells at this time point, GFP+cells were sorted and mixed together with GFP− cells from that tissue prior to cell capture for single-cell RNA sequencing. This resulted in successful capture of 127 E9.0 Nkx2-1$^{GFP+}$ forebrain progenitors (out of a total of 620 cells), but for lung and thyroid progenitors the rarity of GFP progenitors resulted in <3 GFP+ E9.0 cells captured (out of a total of 417 lung and 279 thyroid mixed non-epithelial lineages, respectively). Cells were sorted at the Boston University Flow Cytometry Core Facility using the MoFlo high speed cell sorter and collected in HBSS+buffer. Freshly-sorted cells were spun briefly to concentrate the cells, and then cell concentration and viability of flow-sorted samples were measured using the Countess II instrument (ThermoFisher Scientific) and ranged from 3 to $8 \times 10^5$ cells ml$^{-1}$ with 80% viability. Cell samples were input into 10x Genomics Single Cell 3′ v2 workflow and processed according to the manufacturer's instructions for a targeted recovery of 1000 cells (×10 Genomics, USA). Resulting cDNA libraries were sequenced on an Illumina NextSeq 500 instrument with 1.95 pM input and 1% PhiX control library spike-in (Illumina, USA). Single-cell capture was conducted at the Boston University Single Cell Sequencing Core using a 10X Genomics Chromium System. The resulting reads were aligned and gene level unique molecular identifier (UMI) counts obtain using Cell Ranger version 2.0.1 (×10 Genomics). Four libraries derived from embryonic mouse (two "lung" domain, one "thyroid", and one "forebrain" domain totaling 1316 cells and a post-normalization mean coverage of 165238 reads per cell) were combined for downstream analysis.

Because of the sparse nature of single-cell data, with high dropout levels (undetected but expressed transcripts), we averaged the normalized expression profiles of cells in each cluster and used those averages for the projections. The clusters were identified following the usual procedure[71] of pre-processing and QC filtering, identifying highly variable genes, log-normalization and scaling, dimensionality reduction with PCA and clustering using the Louvain method for community detection at a resolution of 0.5. Clusters were manually annotated based on their marker genes and subsequently averaged, with those averages later exported for downstream analysis.

**Linear algebra projections**. Here, we give a brief overview of the projection method plots shown in Fig. 6 and Supplementary Fig. 7, please see refs. [50,61] for complete mathematical details. The projection method allows one to compare experimentally acquired gene expression data versus a reference basis. A projection score of 1.0 indicates a perfect match of gene expression, while 0.0 indicates no match. The projection method has several key advantages relative to other techniques such as Principal Component Analysis (PCA). First, it provides a biologically interpretable numeric value of the similarity between the data and the reference basis. Second, it removes the inherent correlations of the reference basis by measuring orthogonal projections onto the reference basis. This means that each reference cell type has a projection of 1 with only itself and has a projection of 0 with all other reference cell types.

To create the reference basis cell types, we first started with the basis used in refs. [50,61] (61 cell types). Next, we added data from the following RNA-Seq samples: E8.25 ectoderm, E8.25 foregut endoderm, E9.0 Nkx2-1$^+$ forebrain, E9.0 Nkx2-1$^+$ lung, E13.5 Nkx2-1$^+$ thyroid. We also included single-cell RNA-Seq samples from E9.0 (lung, thyroid, and forebrain domains), and E13.5, E15.5, and E17.5 (lung), averaged over the clusters described above (total of 24 clusters). Finally, we added the average over three E16.5 lung epithelium control samples (GSE57391, sample IDs Ezh2K_01, Ezh2K_11, Ezh2K_15) described in Galvis et al.[51] The basis thus consists of 91 cell types.

The data set consists of a mix of Affymetrix GeneChip Mouse Gene 1.0 ST and RNA-Seq data and was normalized as follows. First, raw microarrays and RNA-Seq were separately processed following standard techniques for each data type. Next, only genes common to all data sets were kept, leading to $N = 10{,}990$ genes. In order to make robust comparisons across platforms, the raw expression output was converted to a rank order. Next, we wanted to convert this rank order to the z-score of a log-normal distribution. We converted the rank to a percentile (for $N$ genes, by dividing by $N+1$), and then this percentile into a normal z-score. For mathematical convenience, we used a biased estimator (i.e., we normalized by $N$ and not $N−1$) since then the Euclidean norm of each microarray gene expression was $N$. At this point, each sample was described by a Gaussian distribution with a Euclidean norm of $N = 10{,}990$. Finally, we selected only the genes that were highly expressed ($z > 1$) in at least one cell type, keeping a total of $N = 9035$ genes for the subsequent analysis. The LAP scores for all conditions are presented in Supplementary Data 3.

As mentioned above, a projection score of 1.0 indicates a perfect match of gene expression, zero indicates no match, and −1.0 indicates perfectly anti-correlated

gene expression. There are no direct equivalents of $P$-values to measure the significance of a projection. However, one can compare the expression of random sets of gene expression versus the reference basis. A random set of 10,000 different gene expression profiles (each with $N = 9035$ genes) had a mean projection of zero (within machine precision) with any of the reference cell types, while the standard deviation of the projections was 0.04. Therefore, a projection greater in magnitude of 0.2 (five standard deviations) is highly unlikely to be due to random expression patterns. This is also supported by the fact that the variation in the projections of each experimental replicate was minimal.

**Gene set analysis**. In order to validate the scores obtained above, we applied an orthogonal signature projection method based on gene set analysis. In particular, we computed separate enrichment scores for each pairing of sample and gene set using the projection methodology described in ref. [72], as implemented in the GSVA package[73].

The procedure can be divided in the following steps: first, we filtered, log-transformed and scaled the data. Second, we selected the genes that constitute the molecular signatures for each of the conditions we want to score: "E8.25 Foregut Endoderm", "E8.25 Ectoderm", "E9.0 Nkx2-1$^+$ Lung", "E9.0 Forebrain", "Liver", "Npc", and "Mef" (for the experimental conditions) and "E8.25 Foregut Endoderm", "E8.25 Ectoderm", "E13.5 Thyroid", "Esc" (for the control conditions). Third, we computed the enrichment score of each of those gene sets for our experimental conditions ("2D-Nkx2-1$^+$", "2D-Nkx2-1$^+$EPCAM$^+$", "3D-Nkx2-1$^+$ EPCAM$^+$", and "3D-Nkx2-1$^-$EPCAM$^+$") and our controls ("D1", "D7", and "D14").

For the first step, we filtered out genes that had a total log-normal expression <2 overall samples in our RNA-Seq data sets. Log transformation and normalization was done using counts per million and a prior count offset to avoid taking log of zero. Subsequently, the values were centered around 0, and the distributions were visualized to evaluate effective normalization and Gaussian shape. Finally, we minimized the effect of outliers by capping extreme values to 1.5 times the interquartile range for each sample.

For the second step, we extracted the 3000 genes with higher variance overall, to capture those with biological signal, and then, for each of the conditions we wanted to score, we extracted the 100 most expressed among those highly variable genes. These are the gene sets to be used for scoring our problem samples in the following step.

In the third and final step, we computed enrichment scores by integrating the difference of the weighted Empirical Cumulative Distribution Function (ECDF) of the genes in the signature and the ECDF of the remaining genes as in the aforementioned publications[72,73].

**Reporting summary**. Further information on research design is available in the Nature Research Reporting Summary linked to this article.

## Data availability
All sequencing data that support the findings of this study have been deposited in the National Center for Biotechnology Information Gene Expression Omnibus (GEO) under to following accession codes: (a) in vivo bulk RNA-Seq: E8.25 Foregut Endoderm and Ectoderm, E9.0 Nkx2-1+Lung and Brain, E9.0 Nkx2-1- Lung and Brain, E13.5 Nkx2-1+ Thyroid, E13.5 Nkx2-1- Thyroid; GEO Series accession code GSE138903. (b) In vitro bulk RNA-Seq: 2D-Nkx2-1+EPCAM+, 3D-Nkx2-1+EPCAM+, 2D-Nkx2-1-EPCAM+; GEO Series accession code GSE138676. (c) In vivo E9.0 single-cell RNA-Seq; GEO Series accession code GSE138904. (d) E13.5 lung Nkx2-1$^{GFP+}$ single-cell data; GEO Series accession code GSE139186. The E15.5 and E17.5 lung Nkx2-1GFP+single-cell data have been previously deposited and are accessible under GEO accession code GSE113320. The microarray data containing the 2D-Nkx2-1+condition have been previously deposited and are accessible under GEO series accession code GSE92916. The bulk RNA-Seq for the thyroid directed differentiation (D1, D7, D14 conditions) have been previously deposited and are accessible under GEO series accession code GSE92572. All other relevant data are available from the corresponding author upon reasonable request.

## Code availability
All custom LAP scripts have been made available at [https://github.com/Emergent-Behaviors-in-Biology/lung-primordium]. Additional modified scripts can be accessed upon request.

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

## Acknowledgements

We thank Drs. Avi Spira and Gang Liu of the BU School of Medicine (BUSM) Computational Biomedicine Section for help with RNA-Sequencing. We are indebted to Dr. Felicia Chen from BUSM Pulmonary Center for help with embryo staging and dissections. We thank Dr. Paul Gadue (UPenn) for the ENDM1 antibody gift. We are grateful to Brian R. Tilton and Dr. Patrick Autissier of the BUSM Flow Cytometry Core for technical assistance, supported by NIH Grant 1UL1TR001430, and Drs. Greg Miller and Marianne James of the CReM, supported by grants R24HL123828 and U01TR001810. We thank Roberto Luis-Fuentes for assistance with epifluorescence microscopy. S.L.L. was supported by a Boston University Clinical and Translational Science Institute (CTSI) training grant (TL1TR001410). M.M.K. is a Marshall Plan Scholar supported by the Austrian Marshall Plan Foundation. R.M. and P.M. were supported by NIH NIGMS grant 1R35GM119461 and a Simons Investigator Award in the Mathematical Modeling of Living Systems (MMLS) to P.M. L.I. was supported by NIH grants (R01 HL111574, R01 HL124280), a chILD Foundation/American Thoracic Society award, an Evans Junior Faculty Research Merit Award and a CTSI Pilot Award (1UL1TR001430). D.N.K. was supported by NIH grants U01HL134745, R01HL095993, R01DK105029, R01GM122096, R01HL122442, R01HL128172, and U01HL134766.

## Author contributions

L.I. made the figures and L.I. and D.N.K. wrote the paper. L.I., M.J.H, S.L.L., R.M.S., K.D., J.H.-B., M.L.L., C.C., A.J.F. and J.M.S. performed experiments. I.S.C. and M.M.K. performed RNA-Seq and GSEA analyses. R.M., C.V.-M. and P.M. conceived and performed computational analyses. D.B.F. and E.E.M. contributed data sets. L.I., M.J.H., J.M.S. and D.N.K. conceived and designed experiments. L.I., E.K., X.V., P.M. and D.N.K. supervised research.

## Competing interests

The authors declare no competing interests.
