## [Peer Review File · Nature Communications]

Reviewers' Comments:

Reviewer #1:

Remarks to the Author:

Ikonomou and colleagues employ an Nkx2.1-GFP mouse strain to measure the transcriptomes of embryonic lung, thyroid, and forebrain cells at an early stage of development, as well as of Nkx2.1-negative foregut endoderm cells. They then identify differentially expressed transcripts for each population and perform GSEA, demonstrating that their analysis rediscovers known signaling pathways important for early lung development, including Wnt and Shh, which they are also able to functionally validate. The authors then compare the efficacy of 2D gelatin versus 3D Matrigel plates for directed differentiation of lung epithelial lineages from PSCs. They determine the 3D condition generates a larger pool of Nkx2.1+ cells and a similarity analysis demonstrates that the cells more closely resemble E17.5 distal epithelial progenitors.

The work is of high quality and the manuscript is very well written and easy to follow. The dataset the authors have generated should be valuable for the lung biology field.

Major points:

- 1) How the finding of "adherens junction" pathway being involved in lung specification informed the 2D gelatin versus 3D Matrigel culture comparison is unclear, as is the predicted result. Do the authors know which is more conducive to formation of adherens junctions, and how do they know the results has anything to do with this property? How much of the effect is due to the 2D versus 3D compared with the gelatin versus Matrigel composition? Is there evidence that the adherens junction pathway was more active in the 3D versus 2D conditions, which would at least support the relevance of the analysis for rationally engineering directed differentiation strategies?
- 2) Some of the micrographs are of insufficient quality or magnification to support the interpretations being claimed. In Figure 1, for instance, the attribution of nuclear GFP to AT1 (Pdpn) and ciliated (acet Tub) cells is not apparent in the pictures provided. In the Scgb1a1 stain, some of the arrows are not even pointing to specific cells.
- 3) A little more discussion or elaboration about the similarity analysis would be useful. It is not clear what is the reference standard for the radar plots, for instance. The 2D and 3D Nkx2.1+EPCAM+ plots look incredibly similar, yet apparently the outcomes of directed differentiation are significantly different between the two conditions. What does one make of these plots in absolute terms, as to the fidelity of the profiles to the actual cells to which they are being compared?
- 4) The authors have not indicated whether and where they will deposit the RNA-seq datasets they have generated. It is essential to make the data publicly available.

Minor points:

- 1) In Figure 1F, there appear to be several Nkx2.1+ but GFP- cells in the field. Is this correct or an immunostaining artifact? If the Nkx2.1-GFP strain is working properly, it seems there should not be any cells with this molecular profile.
- 2) The claim that the image in Supp Figure 1F shows that Nkx2.1-lineage cells are exclusively epithelial at E18.5 based on co-staining for GFP and E-cadherin is totally unconvincing. It does not appear reasonable to make this adjudication based on the image shown. However, the authors do not need to make this claim based on E18.5, since their E14.5 image (where the epithelial identity of the GFP+ cells can be unequivocally assigned) clearly shows this.
- 3) Panel H should be indicated in the legend for Figure 1
- 4) The legends for E and F have been swapped in Figure 2 legend
- 5) There are no scale bars in Figure 1E

Reviewer #2:

Remarks to the Author:

The authors have used computational methods of comparing gene expression profiles to predict culture conditions that modulate directed differentiation of lung epithelial lineages. They tested these predictions experimentally, leading to an more optimized differentiation protocol. Overall, this is exciting work and could serve as more general framework for improving directed differentiation in other systems as well. Although the authors have used innovative analytical methods and the data processing methodology for the new data that have generated appears to be sound, I have the following concerns:

1. I found the way in which the linear algebra projection (LAP) analysis was presented to be somewhat confusing. The authors consistently present the LAP analysis using a series of bar graphs, and it is hard to tell just from looking at the graph which data set is the "query" and which is the "reference" (what is projected onto what?). Also, because there are no x-axis labels on almost any of these graphs, it can be difficult to figure out what is being represented. This is particularly true in Fig. 6B, where it took me some time to realize that the x-axis labels were indicated by the way in which the cartoons of cells in the schematic in Fig. 6A are colored (at least this is what I assume).

2. On a more technical note, the reference basis used for LAP analysis contains three very different data types - microarray expression, RNA-seq, and single-cell RNA-seq. The dynamic ranges of these three measurement types can be quite different from each other, and there are many other technical differences in how these data sets are normalized, processed, etc. I recognize that the authors use non-parametric methods of comparing these data types, which should help to mitigate some of these issues. Nonetheless, it would be worth assessing the extent to which the reference data type influences the "projection strength" computed by LAP analysis. For example, does RNA-seq data project better onto RNA-seq data compared to microarray data?

3. All of the bar graphs from LAP analysis show error bars. However, I couldn't find whether there was a statistical test that accompanies LAP analysis for comparing the projection strengths of different expression profiles. It would be useful to assess the statistical significance of the comparisons of projection strengths. On a related note, it was unclear how many samples are used for computing the average projection strength for each bar in these analyses. If the number of profiles is relatively small, it would probably be worth simply plotting the individual data points (e.g. beeswarm plot or similar).

We would like to thank the reviewers for their insightful and constructive criticism that helped us improve the manuscript. Below, you will find our point-by-point reply to the critiques. Our reply is in *italicized blue font*.

Reviewer #1 (Remarks to the Author):

Ikonomou and colleagues employ an Nkx2.1-GFP mouse strain to measure the transcriptomes of embryonic lung, thyroid, and forebrain cells at an early stage of development, as well as of Nkx2.1-negative foregut endoderm cells. They then identify differentially expressed transcripts for each population and perform GSEA, demonstrating that their analysis rediscovers known signaling pathways important for early lung development, including Wnt and Shh, which they are also able to functionally validate. The authors then compare the efficacy of 2D gelatin versus 3D Matrigel plates for directed differentiation of lung epithelial lineages from PSCs. They determine the 3D condition generates a larger pool of Nkx2.1+ cells and a similarity analysis demonstrates that the cells more closely resemble E17.5 distal epithelial progenitors.

The work is of high quality and the manuscript is very well written and easy to follow. The dataset the authors have generated should be valuable for the lung biology field.

Major points:

1) How the finding of “adherens junction” pathway being involved in lung specification informed the 2D gelatin versus 3D Matrigel culture comparison is unclear, as is the predicted result. Do the authors know which is more conducive to formation of adherens junctions, and how do they know the results has anything to do with this property? How much of the effect is due to the 2D versus 3D compared with the gelatin versus Matrigel composition? Is there evidence that the adherens junction pathway was more active in the 3D versus 2D conditions, which would at least support the relevance of the analysis for rationally engineering directed differentiation strategies?

*We thank the reviewer for raising this important point. Adherens junctions are present in the lung epithelium throughout development (e.g. see (Mahoney et al., 2014)) and E-cadherin expression is a defining feature of such junctions in epithelial cells. To answer the questions raised by the reviewer, we performed a series of experiments and generated new data that are included in the new **Supplementary Figure 5**. We summarize and interpret these data below.*

*The reviewer was right to point out that in the experimental conditions tested (2D Gelatin, old protocol and 3D Matrigel, new protocol) two experimental variables (ECM composition, substratum elastic modulus) were simultaneously changed. Thus we performed lung specification on three substrata (2D Gelatin, 2D Matrigel, and 3D Matrigel) and evaluated E-cadherin expression. As is evident in **Supplementary Figure 5A**, there are very few E-cadherin+ clusters in the “2D Gelatin” condition whereas the highest number of E-cadherin+ clusters occurs in the “3D Matrigel” condition. This correlates well with the percentages of putative lung primordial progenitors (Nkx2-1^{mCherry+}EPCAM⁺) in the respective conditions, indicating that adherens junctions formation is associated with the in vitro generation of such progenitors. We then hypothesized that inhibition of adherens junction formation will adversely affect in vitro lung specification. The data shown in **Supplementary Figures 5B,C** appear to validate this hypothesis as disruption of adherens junctions by incubation with an anti-E-cadherin antibody results in reduction of the Nkx2-1^{mCherry+}EPCAM⁺ population. As no dose escalation for the anti E-cadherin antibody was attempted, it is possible that higher antibody*

concentrations will lead to more drastic effects on the derivation of lung primordial progenitors. Overall, we hope that the new data added to the manuscript strengthen the argument that formation of adherens junctions has an important role in the derivation of an epithelial lung primordial progenitor population that more closely resembles its in vivo counterpart.

2) Some of the micrographs are of insufficient quality or magnification to support the interpretations being claimed. In Figure 1, for instance, the attribution of nuclear GFP to AT1 (Pdpn) and ciliated (acet Tub) cells is not apparent in the pictures provided. In the Scgb1a1 stain, some of the arrows are not even pointing to specific cells.

*We thank the reviewer for this astute observation. The arrows for Scgb1a1 have been rectified in this revised version. We have added magnified views as insets in the figure to make it easier to appreciate nGFP co-staining in PDPN+ cells and acetylated α -tubulin+ cells in **Figure 1C**.*

3) A little more discussion or elaboration about the similarity analysis would be useful. It is not clear what is the reference standard for the radar plots, for instance. The 2D and 3D Nkx2.1+EPCAM+ plots look incredibly similar, yet apparently the outcomes of directed differentiation are significantly different between the two conditions. What does one make of these plots in absolute terms, as to the fidelity of the profiles to the actual cells to which they are being compared?

*The radar plots were generated to visualize the gene set analysis (GSA) which is complementary to the Linear Algebra Projection (LAP) analysis. In the GSA plots, the reference standards are the transcriptomes of the indicated in vivo populations. We have modified the schematic depiction of our computational analysis in **Figure 6A** to clarify what are the “reference lineages” in each type of analysis and what are the in vitro derived lineages that are compared to the former. In the case of 2D and 3D Nkx2-1+EPCAM+ populations these plots as well as the E9.0 projection strength score for the E9.0 in vivo Lung Primordium are indeed very similar. We attribute this to the fact that in either condition we sort out the Nkx2-1+EPCAM+ cells, thus enriching for the putative lung primordial progenitor populations. This is supported by the fact that when one compares the 2D Nkx2-1+ condition (old protocol, no enrichment of epithelial Nkx2-1+ progenitors) to the 2D Nkx2-1+EPCAM+ condition (old protocol, highly enriched epithelial Nkx2-1+ progenitors) LAP projection scores for E9.0 in vivo Lung Primordium and mesenchymal cell types as well as GSA radar plots are markedly different. Therefore, although lung specification outcomes in terms of epithelial progenitor yields are different between 2D and 3D conditions, Nkx2-1+ epithelial progenitors derived in either conditions are quite similar and most of the lung competence in the 2D condition appears to be in the small (~1-2%) Nkx2-1+EPCAM+ population. We added a sentence to the Discussion to further emphasize this point (Page 28 of the manuscript)*

4) The authors have not indicated whether and where they will deposit the RNA-seq datasets they have generated. It is essential to make the data publicly available.

We have deposited to GEO all the RNA-Seq datasets we generated for this project. We have set an embargo date for the newly deposited data and they will become publicly available upon the acceptance of our manuscript and we have added a statement with this information in the Methods section (page 45). Nevertheless, we have created access tokens that can be used by the reviewers to access the data, should they wish.

More specifically, the following datasets were deposited:

- a) *In vivo bulk RNA-Seq: E8.25 Foregut Endoderm and Ectoderm, E9.0 Nkx2-1+ Lung and Brain, E9.0 Nkx2-1- Lung and Brain, E13.5 Nkx2-1+ Thyroid, E13.5 Nkx2-1- Thyroid; GEO Series accession number GSE138903*

To review GEO accession GSE138903: Go to
<https://www.ncbi.nlm.nih.gov/geo/query/acc.cgi?acc=GSE138903>. Enter token
szitcwmqbfshjol into the box

b) *In vitro* bulk RNA-Seq; 2D-Nkx2-1+EPCAM+, 3D-Nkx2-1+EPCAM+, 2D-Nkx2-1-
EPCAM+; GEO Series accession number GSE138676
To review GEO accession GSE138676: Go to
<https://www.ncbi.nlm.nih.gov/geo/query/acc.cgi?acc=GSE138676>. Enter token
clqzsgmglzsfpez into the box

c) *In vivo* E9.0 single-cell RNA-Seq; GEO Series accession number GSE138904
To review GEO accession GSE138904: Go to
<https://www.ncbi.nlm.nih.gov/geo/query/acc.cgi?acc=GSE138904>, Enter token
kbsdysuobfshjcp into the box

d) E13.5 lung Nkx2-1^{GFP+} single-cell data; GEO Series accession number XXX

The single cell RNA-Seq from E15.5 and E17.5 Nkx2-1^{GFP+} lung cells had been previously
deposited (GEO accession number GSE113320:
<https://www.ncbi.nlm.nih.gov/geo/query/acc.cgi?acc=GSE113320>)

Minor points:

1) In Figure 1F, there appear to be several Nkx2.1+ but GFP- cells in the field. Is this correct or an immunostaining artifact? If the Nkx2.1-GFP strain is working properly, it seems there should not be any cells with this molecular profile.

This is indeed correct and we have repeated the staining. A better quality confocal micrograph showing GFP+ cells co-expressing nuclear NKX2-1 protein has replaced the old one.

2) The claim that the image in Supp Figure 1F shows that Nkx2.1-lineage cells are exclusively epithelial at E18.5 based on co-staining for GFP and E-cadherin is totally unconvincing. It does not appear reasonable to make this adjudication based on the image shown. However, the authors do not need to make this claim based on E18.5, since their E14.5 image (where the epithelial identity of the GFP+ cells can be unequivocally assigned) clearly shows this.

*We agree with the reviewer that the exclusively epithelial identity of the Nkx2-1 lineage-traced cells is well established based on the E14.5 micrographs. Therefore we decided to remove **Supplementary Figure 1F** to avoid confusing the reader.*

3) Panel H should be indicated in the legend for Figure 1

*We have indicated the content of Panel H in the **Figure 1** legend.*

4) The legends for E and F have been swapped in Figure 2 legend

We have corrected this and highlighted the change.

5) There are no scale bars in Figure 1E

*Micrographs in **Figure 1E** were obtained using an epifluorescence stereomicroscope at various magnifications for the main purpose of demonstrating the localization of GFP during lung*

embryonic development. Unfortunately no scale measurements were recorded at the time of capture and therefore we cannot provide scale bars for each micrograph.

Reviewer #2 (Remarks to the Author):

The authors have used computational methods of comparing gene expression profiles to predict culture conditions that modulate directed differentiation of lung epithelial lineages. They tested these predictions experimentally, leading to an more optimized differentiation protocol. Overall, this is exciting work and could serve as more general framework for improving directed differentiation in other systems as well. Although the authors have used innovative analytical methods and the data processing methodology for the new data that have generated appears to be sound, I have the following concerns:

1. I found the way in which the linear algebra projection (LAP) analysis was presented to be somewhat confusing. The authors consistently present the LAP analysis using a series of bar graphs, and it is hard to tell just from looking at the graph which data set is the "query" and which is the "reference" (what is projected onto what?). Also, because there are no x-axis labels on almost any of these graphs, it can be difficult to figure out what is being represented. This is particularly true in Fig. 6B, where it took me some time to realize that the x-axis labels were indicated by the way in which the cartoons of cells in the schematic in Fig. 6A are colored (at least this is what I assume).

*We modified **Figure 6** to make understanding of the "query" and "reference" datasets easier. In the new **Figure 6A** that summarizes our computational approach, we clearly indicate for each method (Linear Algebra Projections (LAP) or Gene Set Analysis (GSA)) the "query" dataset (In Vitro Derived Lineages) that are also color-coded and the "reference" datasets (Reference Lineages). We hope the improved schematic outline of our approach makes the subsequent subfigures more legible.*

2. On a more technical note, the reference basis used for LAP analysis contains three very different data types - microarray expression, RNA-seq, and single-cell RNA-seq. The dynamic ranges of these three measurement types can be quite different from each other, and there are many other technical differences in how these data sets are normalized, processed, etc. I recognize that the authors use non-parametric methods of comparing these data types, which should help to mitigate some of these issues. Nonetheless, it would be worth assessing the extent to which the reference data type influences the "projection strength" computed by LAP analysis. For example, does RNA-seq data project better onto RNA-seq data compared to microarray data?

We thank the reviewer for this very important question. We have spent a lot of time thinking about this issue and are currently writing a manuscript explaining these questions (figures from this manuscript are included below). The short answer is that we have found that by converting measurements to z-scores we can mitigate many of these problems. We have included a statement in the main manuscript to emphasize this point (page 19 of the manuscript) and we provide a more technical explanation below.

As explained in the methods, in brief we first rank-order genes, convert this rank ordering to a percentile, which in turn gives rise to a z-score. We call this procedure RankNorm. We have found that this procedure allows for robust comparisons across platform using LAPs.

We include here two figures from the manuscript in preparation detailing these procedures. In the first figure, we compare how expression correlates across platforms. The figure compares three platforms (Affymetrix U133A, Affymetrix Exon 1.0 ST, and RNA Seq) using raw log₂ data, rank ordering, and Z-scores. As can be clearly seen, the z-scoring improves the capacity to compare expression levels across platforms, whereas raw scores are only moderately comparable.

In the second figure, we assess the robustness of the LAP scores across platforms. To do so, we used the expression profiles of ESC and iPSCs from multiple labs across all three platforms in both mouse (most relevant for this study) and humans. Panels C,D show that performing a principal component analysis (PCA) of all these expression profiles largely segregates by technology. Such batch effects are known to be large in PCA based techniques. In panels E and F, we have tested the LAP method. To do so, we created an “ESC basis vector” consisting of the median expression for all genes across all ESC cell lines in our dataset. We then calculated the LAP scores for various ESC and iPSC cell lines. As can be seen clearly, the LAP score is extremely consistent and high ~0.7-0.8. We also calculated the maximum projection on any of the other ~70 basis vectors corresponding to other cell types and found that they were close to 0.1. We emphasize that the ESCs and iPSCs in these datasets were generated from many different cell types, with different protocols, and grown in different media conditions, indicating that our method is quite robust to large variations (though the outliers in Panel D almost all come from one lab that has different culturing protocols distinct from others).

Figure 1. Technology and normalization comparisons for RankNorm. The samples correspond to the median Spearman correlation (see SI Fig 5 for details) for each comparison between technologies and normalizations. The first column (A, D, G) shows Affymetrix U133A vs Affymetrix Exon 1.0 ST. The second column (B, E, H) shows Affymetrix U133A vs RNA-Seq normalized by RSEM. The third column (C, F, I) shows Affymetrix Exon 1.0 ST vs RNA-Seq normalized by RSEM. The first row (A, B, C) is the raw log₂ data expression with the lowest expression set to 0. The second row (D, E, F) is the rank order expression. The third row (G, H, I) is the Z-score expression after applying RankNorm.

Figure 3. ESC vs iPSC comparison. Both mouse and human ESC and iPSC from multiple labs, protocols, and gene expression technologies are compared. The first column (A, C, E) is mouse data while the second column (B, D, F) is human data. A and B are respectively the legends for the mouse and human data. Basis stands for the representative expression for different cell types. The microarray technologies are all Affymetrix. C and D are the Principal Component Analysis (PCA) of ESC, iPSC, and other cell types for respectively mouse and human. E and F show ESC vs iPSC when analyzed using Linear Algebra Projections (LAP) respectively for mouse and human. The plots show the projection onto the typical ESC vs the maximum projection onto any other basis cell type.

3. All of the bar graphs from LAP analysis show error bars. However, I couldn't find whether there was a statistical test that accompanies LAP analysis for comparing the projection strengths of different expression profiles. It would be useful to assess the statistical significance of the comparisons of projection strengths. On a related note, it was unclear how many samples are used for computing the average projection strength for each bar in these analyses. If the number of profiles is relatively small, it would probably be worth simply plotting the individual data points (e.g. beeswarm plot or similar).

We thank the reviewer for this comment. The error bars in the old plots were standard deviations over three replicates mentioned in the methods. We have removed the error bars and instead show each of the LAP scores separately. As can be seen, the LAP scores are consistent across replicates.

To understand if a LAP score is significant, as noted in the methods, a random vector has a LAP score with mean 0 and standard deviation of 0.04. This provides a natural scale with which to compare LAP scores. This is explained in the methods as follows:

“A random set of 10,000 different gene expression profiles (each with N=9035 genes) had a mean projection of zero (within machine precision) with any of the reference cell types, while the standard deviation of the projections was 0.04. Therefore, a projection greater in magnitude of 0.2 (five standard deviations) is highly unlikely to be due to random expression patterns. This is also supported by the fact that the variation in the projections of each experimental replicate was minimal.”

Comments from Referee #3:

This study examines the genetic program that drive primordial lung progenitors. The authors show that Nkx2-1GFP line faithfully traces lung epithelial cells in various developmental stages. From there, they also show that E9.0 forebrain, E9.0 lung and E13.5 thyroid possess genetic regulatory network of their own, though these organs all require Nkx2-1 at first place. Further bioinformatic analysis reveals differences, both among organs and from precursors to progenitors. Notably, the authors validate the role of Wnt and FGF signaling in vitro. Finally, a linear algebra projection analysis was carried out on 2D/3D cultured lung epithelial progenitors and revealed importance of microenvironment as well as biomechanical forces. This work dissects the underlying determinants for primordial lung progenitors both in vitro and in silico. Overall, the manuscript presents a rigorous logic and comprehensive data which should raise interests from readers.

Specific points:

1) In Figure 2F, lung Nkx2-1- population has a similar genetic program compared to forebrain populations (both Nkx2-1+ and Nkx2-1-). However, this population is quite distinct from lung Nkx2-1+ population based on PC1 and PC2. What would be the interpretation?

Indeed, the Nkx2-1- population that was isolated from the lung field is very close to the brain populations along the PC2 axis and appears to partially overlap with the Nkx2-1- brain populations. We want to clarify that the Nkx2-1- “brain” population is not a forebrain population but it contains all Nkx2-1- lineages that are isolated after digestion of the embryo head at E9.0. In an analogous way, the Nkx2-1- “lung field” population is a heterogeneous population that

most probably contains mesenchymal, endothelial, neuronal, non-lung foregut, and other lineages. The fact that both Nkx2-1- populations are highly heterogeneous may account for their similarity as reflected in their partial overlap along the PC2 axis. We have edited the legend of Figure 2 to make this point more clear.

2) 2D-Nkx2-1+EPCAM+ and 3D-Nkx2-1+EPCAM+ seemed to recapitulate different stages of embryonic lung. Would culture time affect the results, i.e. would 3D-Nkx2-1+EPCAM+ be more similar to E13.5 lung cells if analyzed at an earlier time?

This is an interesting point and we do not have a more definitive answer at this time. As we allude to in the discussion, this difference in projection scores for later developmental time points may be due to fate biases introduced by the different ECM substrata, the presence of distinct subpopulations or both. Future single-cell RNA-Seq studies of in vitro ESC-derived lung progenitors that are outside the scope of this manuscript may offer answers to these questions.

3) In Figure 1H, some of the columns don't have error bar.

We apologize for the fact that the error bars were not visible in the "NKX2-1⁻EPCAM⁺" condition (dark blue bars). We changed the color of the errors bars from black to white for this condition to make them visible.

4) In Figure 2A, show the population percentage in foregut endoderm gating panel.

We added the respective information to the flow cytometry plot.

5) Line 197-198: "On average, forebrain cells expressed higher levels of Nkx2-1 transcripts compared to lung and thyroid at E9.0" The thyroid is from E13.5 instead of E9.0.

We thank the reviewer for the observation, we have corrected this sentence.

References

Mahoney, J.E., Mori, M., Szymaniak, A.D., Varelas, X., and Cardoso, W.V. (2014). The Hippo Pathway Effector Yap Controls Patterning and Differentiation of Airway Epithelial Progenitors. *Dev Cell* 30, 137-150.

Reviewers' Comments:

Reviewer #1:

Remarks to the Author:

The authors have done a nice job of addressing my original criticisms. They have performed an additional culture experiment to tease apart which component of the conditions appears to confer the enhanced effect on lung cell differentiation. They have also added in higher quality micrographs to illustrate their immunostaining colocalization claims.

Overall, the manuscript is much improved and should be a valuable contribution to the lung biology field.

Reviewer #2:

Remarks to the Author:

I had no concerns about the significance of this work in my previous review, and I remain enthusiastic. With respect to my criticisms of the clarity of the figures and statistical analysis, I find the responses and modifications made by the authors to be satisfactory.

We would like to thank the reviewers for their kind comments. We are glad the revised version of the manuscript and our detailed reply were deemed satisfactory by all the reviewers.

REVIEWERS' COMMENTS:

Reviewer #1 (Remarks to the Author):

The authors have done a nice job of addressing my original criticisms. They have performed an additional culture experiment to tease apart which component of the conditions appears to confer the enhanced effect on lung cell differentiation. They have also added in higher quality micrographs to illustrate their immunostaining colocalization claims.

Overall, the manuscript is much improved and should be a valuable contribution to the lung biology field.

--

Reviewer #2 (Remarks to the Author):

I had no concerns about the significance of this work in my previous review, and I remain enthusiastic. With respect to my criticisms of the clarity of the figures and statistical analysis, I find the responses and modifications made by the authors to be satisfactory.